

# Experimental and computational kinetics investigations for the reactions of Cl atoms with series of unsaturated ketones in gas phase

**Siripina Vijayakumar, Avinash Kumar, and Balla Rajakumar***

 Department of Chemistry, Indian Institute of Technology Madras, Chennai 600036, India.

*Address for correspondence: rajakumar@iitm.ac.in

http://chem.iitm.ac.in/faculty/rajakumar/

**Abstract**

Temperature dependent rate coefficients for the gas phase reactions of Cl atoms with 4-hexen-3-one and 5-hexen-2-one were measured over the temperature range of 298-363K relative to 1-pentene, 1,3-butadiene and isoprene. Gas Chromatography (GC) was used to measure the concentrations of the organics. The derived temperature dependent Arrhenius expressions are $k_{4\text{-hexen-3-one+Cl (298-363K)}}$ = $(2.82\pm1.76)\times10^{-12}\exp[(1556\pm438)/T]$ $cm^3molecule^{-1}s^{-1}$ and $k_{5\text{-hexen-2-one+Cl (298-363K)}}$ = $(4.6\pm2.4)\times10^{-11}\exp[(646\pm171)/T]$ $cm^3molecule^{-1}s^{-1}$. The corresponding room temperature rate coefficients are $(5.54\pm0.41)\times10^{-10}$ $cm^3molecule^{-1}s^{-1}$ and $(4.00\pm0.37)\times10^{-10}$ $cm^3molecule^{-1}s^{-1}$ for the reactions of Cl atoms with 4-hexen-3-one and 5-hexen-2-one respectively. To understand the mechanism of Cl atom reactions with unsaturated ketones, computational calculations were performed for the reactions of Cl atoms with 4-hexen-3-one, 5-hexen-2-one and 3-penten-2-one over the temperature range of 275-400K using Canonical Variational Transition state theory (CVT) with Small Curvature Tunneling (SCT) in combination with CCSD(T)/6-31+G(d, p)//MP2/6-311++G(d, p) level of theory. Atmospheric implications, reaction mechanism and feasibility of the title reactions are discussed in this manuscript.

**Key words**

Unsaturated ketones, kinetics with Cl atoms, Canonical Variational Transition state theory (CVT) with Small Curvature Tunneling (SCT), Temperature dependent rate coefficients, Atmospheric life times.





## 1. Introduction

In recent past, large amounts of hydrocarbons such as ketones, aldehydes, ethers, esters, alcohols, acids and aromatic compounds are being released into the atmosphere as they have wide usage in manufacturing of plastics, electronic components, polymers, textile and other industrial applications (Graedel et al., 1978). These hydrocarbons are removed from the atmosphere by several physical and chemical processes such as wet deposition, chemical degradation and photolysis. Further reactions of these compounds leads to the formation of secondary organic aerosols (SOAs) and photochemical smog (Derwent et al., 1996 and Hatakeyama et al., 1987) which effects the air quality, climate change and human health. Recent studies show that the global oxidation of biogenic volatile organic compounds contributing 90% of the SOAs in the atmosphere (Kanakidou et al., 2005).

Among all the hydrocarbons, ketones have received an abundant attention as they play an important role in the Earth's atmosphere. They form free radicals on reacting with atmospheric oxidizing species such as OH radicals, Cl atoms, $NO_3$ radicals and $O_3$ molecules. These free radicals are the important intermediates in the formation of aerosols and main sources for other oxidants such as ozone, peroxycarboxylic acids and peroxyacyl nitrates which can affect the oxidation capacity of the atmosphere (Calvert et al., 1987). In combustion chemistry, these ketones play an important role as they are formed as intermediates during the oxidation of oxygenated fuels and hydrocarbons at high temperatures (Rothamer et al., 2009). The end products of these carbonyl compounds are formaldehyde, acetaldehyde and acetone which contribute 40% of the ozone formation in polluted urban areas (Duan et al., 2008).

4-hexen-3-one, 5-hexen-2-one and 3-penten-2-one are being released into Earth's atmosphere by variety of plants from its leaves and by cooking waste (Karl et al., 2001; Tanchotikul et al., 1989 and Shimoda et al., 1995). These compounds are widely used in manufacturing of insecticides, fungicides and as solvents. Combustion of residential wood, burning of biomass and vegetation are the significant sources for releasing 4-hexen-3-one into the atmosphere. 4-hexen-3-one is an intermediate product in the synthesis of variety of perfumes (Wiley-VCH, 2011). As these compounds are released excessively via both anthropogenic and natural sources, they might exhibit certain impacts on the Earth's atmosphere. Therefore, as a first step, to understand the fate of these molecules in the troposphere, one should know the rate coefficients for their





reactions with different atmospheric oxidants. In this attempt, Wang et al., 2010 measured the
rate coefficients for the reactions of 5-hexen-2-one with OH radicals, $O_3$ molecules, and $NO_3$
radicals and reported them to be $k_{298K}=(4.49\pm1.02)\times10^{-11}$ $cm^3molecule^{-1}s^{-1}$,
$k_{298K}=(9.17\pm0.15)\times10^{-18}$ $cm^3molecule^{-1}s^{-1}$ and $k_{298K}=(2.16\pm0.21)\times10^{-14}$ $cm^3$ $molecule^{-1}s^{-1}$
respectively. Blanco et al., 2012 measured the rate coefficients for the reactions of Cl atoms with
4-hexen-3-one, 5-hexen-2-one and 3-penten-2-one at room temperature and one atmospheric
pressure and reported them to be $k_{298K}=$ $(3.00\pm0.58)\times10^{-10}$ $cm^3molecule^{-1}s^{-1}$,
$k_{298K}=(3.15\pm0.50)\times10^{-10}$ $cm^3molecule^{-1}s^{-1}$ and $k_{298K}=(2.53\pm0.54)$ $\times10^{-10}$ $cm^3molecule^{-1}s^{-1}$
respectively. Blanco et al.,[13] also measured the rate coefficients for the reactions of OH radicals
with 4-hexen-3-one, 5-hexen-2-one and 3-penten-2-one and reported them to be k=
$(9.04\pm2.12)\times10^{-11}$ $cm^3molecule^{-1}s^{-1}$, $k=(5.18\pm1.27)\times10^{-11}$ $cm^3molecule^{-1}s^{-1}$ and
$k=(7.22\pm1.74)\times10^{-11}$ $cm^3molecule^{-1}s^{-1}$ respectively at 298K and atmospheric pressure.
Temperature and pressure dependent rate coefficients for the reactions of test molecules with Cl
atoms are required to understand their atmospheric fate in urban polluted areas and marine
boundary layers where the Cl atom concentrations are very high.
Till date, no data is available on the temperature dependence of the rate coefficients for the test
molecules. In this study, the temperature dependent rate coefficients of the proposed reactions
were measured. The global rate coefficients can be measured experimentally but theoretically the
individual rate coefficients can be calculated for each and every reaction site, the complete
potential energy surface i.e. how the reaction takes place from the reactant to product via
different transition states. Thermodynamic parameters were calculated to know the feasibility of
the reactions. Hence, computational calculations were performed for the reactions of Cl atoms
with 4-hexen-3-one, 5-hexen-2-one and 3-penten-2-one using CVT/SCT in combination with
CCSD(T)/6-31+G(d,p)//MP2/6-311++G(d,p) level of theory.





## 2.1. Experimental Methodology

Temperature dependent rate coefficients were measured for the title reactions using relative rate
experimental technique. The complete experimental set up used in this study was described in
our previous article (Vijayakumar et al., 2016). Experiments were carried out in a double walled
Pyrex reaction chamber of 1250 $cm^3$ volume. Cl atoms were produced in situ by photolysis of
oxalylchloride $(COCl)_2$ at 248 nm using an Excimer laser (Coherent Compex Pro). The
experiments were carried out with a laser fluence of 5-6 mJ pulse$^{-1}$. Reaction mixture was
photolyzed by 1200, 1400, 1600, 1800 and 2000 pulses and allowed for 20 minutes to equilibrate
before analysis. Gas Chromatograph (GC, Agilent Technologies 7890B) coupled with Flame
Ionization detector (FID) was used to measure the concentrations of test molecules and reference
compounds. The GC was operated at optimized temperature and flow conditions to elute the test
molecules with distinguishable retention times. HP plot Q column (30m× 0.320mm×20.0 µm,
19091P-Q04) was used for the analyses in the GC. 1-pentene, isoprene and 1,3-butadiene
compounds were used as reference compounds. The temperature dependent rate coefficients for
the reactions of 1-pentene (Coquet et al., 2000) and isoprene (Bedjanian et al., 1998) with Cl
atom are available in the literature (*vide infra*) and therefore, they were used as reference
compounds in the entire studied temperature range. However, only room temperature (298K,
*vide infra*) rate coefficient is available in case of the reaction of 1,3-butadiene (Notario et al.,
1997) with Cl atom and hence it was used in our studies as reference compound at 298 K alone.
Before the reaction kinetics were investigated, some preliminary tests were performed to
estimate the influence of wall effects. The reaction mixture containing the test molecule,
reference compound and the precursor for Cl atom was kept for 6 hours in dark which is more





than the actual reaction time. The samples were analyzed in the GC at every half-an-hour and
verified for any significant loss of the reactants and no such influence was observed. The sample
mixture without the precursor was irradiated at 248 nm for 5 minutes, to verify the loss of the
compounds due to direct photolysis and no loss was observed. To account for the influence of
any secondary reactions, oxygen was added to the reaction mixture and the experiments were
carried out at room temperature and at extreme temperatures. No change in the measured rate
coefficients was observed for all the test reactions which shows the absence or negligible
influence of secondary reactions due to the radicals formed in the test reaction.
Temperature dependent rate coefficients were measured from the decrease in concentrations of
sample and reference compounds due to their reactions with Cl atoms using the standard relative
rate expression given below.

$$\ln\left\{\frac{[sample]_0}{[sample]_t}\right\} = \frac{k_{sample}}{k_{reference}}\ln\left\{\frac{[reference]_0}{[reference]_t}\right\}$$

where $[sample]_0$, $[sample]_t$, $[reference]_0$ and $[reference]_t$ are the concentrations of the sample and
reference compounds (obtained in GC analyses) at time '0' and 't' respectively. $k_{sample}$ and
$k_{reference}$ are the rate coefficients for their reactions with Cl atoms. The ratios of the rate
coefficients $k_{sample}/k_{reference}$ were obtained by plotting $\ln([sample]_0/[sample]_t)$ versus
$\ln([reference]_0/[reference]_t)$. Rate coefficients for title reactions were obtained by multiplying
the slope with rate coefficient of the reference reaction at the same temperature.
**Chemicals**
1,3-butadiene (purity 99.5%, Praxair), 1-pentene (purity ≥98.5%, Aldrich), isoprene (purity 99%,
Aldrich), 4-hexen-3-one (>90% predominantly trans, Aldrich), 5-hexene-2-one (Aldrich, 99%),
oxalylchloride (purity 98%, Spectrochem), Nitrogen (99.995%, Indogas, Chennai), Oxygen
(98%, Indo gas, Chennai). These compounds were further purified using freeze-pump-thaw
method.





**2.2. Computational Methodology**
The geometries of reactants (4-hexen-3-one, 5-hexene-2-one and 3-penten-2-one), reactive
complexes (RCs), transition states (TSs) and products (Ps) were optimized at MP2 (Moller et al.,
1934) level of theory with different Pople's basis (Frisch et al., 1984) sets such as 6-31+G(d, p)
and 6-311++G(d, p). Intrinsic reaction coordinates (IRCs) calculations for every reaction were
performed at MP2/6-311++G(d, p) level of theory. Transition states were identified with one
negative imaginary frequency and reactants, pre-reactive complexes and products were identified
with zero imaginary frequencies. Single point energy calculations (Rodriguez et al., 2010) were
carried out to get an accurate energy barrier heights at CCSD(T)/6-31+G(d, p) level of theory.
All electronic structure calculations were performed using the Gaussian 09 programme suite
(Frisch et al., 2010).
**2.3. Kinetics**
Temperature dependent rate coefficients were computed for the title reactions over the
temperature range of 275-400K using CVT with SCT (Curtiss et al., 1999 and Gonzalez-Lafont
et al., 1991) at CCSD(T)/6-31+G(d, p)//MP2/6-311++G(d, p) level of theory.

$$k^{GT}(T,s) = \sigma \frac{k_B T}{h} \left( \frac{Q^{GT}(T,S)}{\phi^R(T)} \right) \exp\left( \frac{-V_{MEP}(s)}{k_B T} \right)$$

$$k^{CVT}(T) = min_s k^{GT}(T,s) = k^{GT}[T, s^{CVT}(T)]$$

where $k^{CVT}$ is the rate coefficient calculated using CVT and $k^{GT}$ is the generalized rate
coefficient, h= Planck's constant, σ is the reaction path degeneracy, T= temperature (in Kelvin),
$k_B$ is the Boltzmann constant, $\phi^R$ and $Q^{GT}$ are the partition functions of a generalized reactant at
's' and transition state respectively. $S^{CVT}$ is the reaction coordinates of the canonical variational
transition state dividing surface. $V_{MEP}(s)$ is potential energy of generalized TS at 's'. The
tunneling corrected rate coefficients ($k^{CVT/SCT}$) were obtained by multiplying $k^{CVT}$ and a
temperature dependent transmission coefficient $K^{CVT/SCT}(T)$.

$$k^{CVT/SCT}(T) = K^{CVT/SCT}(T) k^{CVT}(T)$$





**3. Results and Discussion**
**3.1. Reaction of Cl atoms with 4-hexen-3-one**
**3.1.1. Experimental**
Temperature dependent rate coefficients for the reaction of Cl atoms with 4-hexen-3-one were
measured over the temperature range of 298-363K relative to 1-pentene (Coquet et al., 2000),
isoprene (Bedjanian et al., 1998) and 1,3-butadiene (Notario et al., 1997). The rate coefficient for
the reaction of Cl atoms with 1,3-butadiene was used in present investigation which was
measured by Notario et al. by using laser photolysis-resonance fluorescence technique and
reported it to be k= (3.48 ±0.10) ×10$^{-10}$ cm$^3$molecule$^{-1}$s$^{-1}$ at 298K. Bedjanian et al. have reported
the temperature dependent rate coefficient for the reaction of isoprene with Cl atom in the
temperature range of 233-320K and at atmospheric pressure. These rate coefficient were
measured using discharge flow – mass spectrometric method and reported it to be k$_{233-320K}$=
(6.7±2.0)×10$^{-11}$ exp[(485±85)/T] cm$^3$molecule$^{-1}$s$^{-1}$. Coquet et al. have investigated the
temperature dependent rate coefficient for the reaction of 1-pentene with Cl atom in the
temperature range of 283-323K and at atmospheric pressure. These rate coefficients were
measured with reference to n-hexane using relative rate method and reported it to be, k$_{283-323K}$=
(4.0±2.2)×10$^{-11}$ exp[(733±288)/T]cm$^3$ molecule$^{-1}$s$^{-1}$. These rate coefficients were used in the
present investigation. As the temperature dependent rate coefficients for 1,3-butadiene are not
available, the measurements using it were restricted to the room temperature (298K) only.
However, isoprene and 1-pentene were used as reference compounds in the entire studied
temperature range. Rate coefficients were measured at 298, 310, 330, 350 and 363K. The plot of
relative decrease in the concentration of 4-hexen-3-one due to its reaction with Cl atoms relative
to 1-pentene, 1,3-butadiene and isoprene is shown in Figure 1. The behavior of the relative
decrease of the reactant was found to be linear through origin, explains the non-interference of
the secondary chemistry on the reaction. The rate coefficients measured for the reaction of 4-
hexen-3-one are given in Table 1. At room temperature, the obtained rate coefficients for the
reaction R1 relative to 1-pentene, 1,3-butadiene and isoprene are very close to each other
(maximum difference found was about 10%) and therefore, they were averaged and obtained the
rate coefficient $k_{R1-298K}^{Expt}$=(5.54±0.41)×10$^{-10}$ cm$^3$molecule$^{-1}$s$^{-1}$. This rate coefficient at 298K is




45% larger than the theoretically calculated rate coefficient ( $k_{R1-298K}^{Theory}$ = 3.66×10$^{-10}$ cm$^3$molecule$^-$
$^1$s$^{-1}$) *vide infra* and, also the one reported by Blanco et al. at 298K, (3.0±0.5)×10$^{-10}$ cm$^3$molecule$^-$
$^1$s$^{-1}$. May be this is due to the differences in the rate coefficients of the reference compounds and
uncertainties associated with the reference compounds which were used in the present
measurements. Several groups have measured the rate coefficients of Cl atom reactions with
unsaturated hydrocarbons at room temperature (298K), but only two temperature dependent rate
coefficients (Cl + 1-pentene (Coquet et al., 2000) and Cl + isoprene (Bedjanian et al., 1998)) are
available in literature whose rate coefficients are close to the rate coefficients of the title reaction
at the temperatures across the study. Hence these two reference compounds were used in the
present investigation. The obtained rate coefficients relative to 1-pentene and isoprene were
averaged across the studied temperature range and depicted in Table 1. The obtained averaged
rate coefficients were used to fit the temperature dependence by linear least squares method and
is given in Figure 2. The derived temperature dependent Arrhenius expression for the reaction
R1 is $k_{R1-(298-363K)}^{Expt}$ = (2.82±1.76)×10$^{-12}$exp[(1556±438)/T] cm$^3$ molecule$^{-1}$s$^{-1}$.
**3.1.2. Computational**

<p style="text-align:center">TS2a</p>

**Structure 1: 4-hexen-3-one**

Geometries of the reactant, pre-reactive complexes, transition states and products were optimized
at MP2/6-311++G (d, p) level of theory and shown in Figure 3. In the reaction of Cl atoms with
4-hexen-3-one, twelve (two addition channels and ten abstraction channels) transition states were
identified. Addition transition states were represented by TS1a (adjacent to the –CH$_3$ group) and
TS2a (adjacent to the –C=O group) and hydrogen abstraction channels were represented by TS1



to TS10 as shown in the structure 1. The Cl atom addition on double bond at TS1a and TS2a lead
to the formation of secondary alkyl radicals Pla and P2a respectively. In hydrogen abstraction
reactions, the –C-H bond length was elongated from 16% to 40% when compared with the
normal –C-H bond length in the test molecule. The energies of all the stationary points were
refined at CCSD(T)/6-31+G(d, p) using the optimized geometries obtained at MP2/6-311++ G(d,
p) level of theory. Thus refined energies were used in dual level CVT calculations to compute
the rate coefficients for the reaction over the temperature range of 275-400K. The potential
energy diagram for reaction R1 is given in the Figure 4. From this Figure, it is very clear that the
addition transition states (TS1a and TS2a) are submerged and therefore, these two channels
would contribute maximum to the global rate coefficient. The pre-reactive complexes for these
submerged channels were identified and used in the calculations. The global rate coefficient for
the reaction R1 was calculated by linear summation of the individual rate coefficients viz.
$\quad k_{global} = k_{TS1a} + k_{TS2a} + k_{TS1} + k_{TS2} + k_{TS3} + k_{TS4} + k_{TS5} + k_{TS6} + k_{TS7} + k_{TS8} + k_{TS9} + k_{TS10}$
The rate coefficient for the reaction R1 at 298K was calculated to be $k_{R1-298K}^{Theory} = 3.66 \times 10^{-10}$
$cm^3 molecule^{-1} s^{-1}$. It should be noted that the theoretically calculated rate coefficient is in
excellent agreement with the one reported by Blanco et al. at 298K, $(3.0 \pm 0.5) \times 10^{-10}$
$cm^3 molecule^{-1} s^{-1}$. The computed rate coefficients were used to fit the Arrhenius equation by
linear least squares method and is given in Figure 2 along with the experimentally measured
ones. The theoretically obtained temperature dependent rate coefficient for the reaction R1 is
$k_{R1-(275-400K)}^{Theory} = (1.41 \pm 0.13) \times 10^{-12} exp[(1663 \pm 57)/T] \, cm^3 molecule^{-1} s^{-1}$. The activation energies
predicted by both theory and experiment seems to be very close to each other. However, the
experimentally obtained pre-exponential factor is twice that of the one estimated by theory.
Obviously, the trends of the rate coefficients are identical but the intercept, which is essentially
the pre-exponential factor.
**3.2. Reaction of Cl atoms with 5-hexene-2-one**
**3.2.1. Experimental**
Rate coefficients were measured for the reaction of Cl atoms with 5-hexene-2-one over the
temperature range of 298-363K relative to 1-pentene (Coquet et al., 2000) and isoprene





(Bedjanian et al., 1998). The plot of relative decrease in the concentration of 5-hexen-2-one due
to its reaction with Cl atoms relative to 1-pentene and isoprene is shown in Figure 5.  The linear
behavior of the loss through the origin ascertains the non-influence of the secondary/radical
chemistry. This was further checked by carrying out the reaction in the presence of excess
oxygen at extreme temperatures of this study and no change in the measured rate coefficients
was noticed. For the present reaction R2, the rate coefficients were measured at 298, 310, 330,
350 and 363K and they are tabulated in Table 4. The rate coefficients measured at 298 K with 1-
pentene and isoprene as reference compounds differ by 12% and at 363K they are found to be
same. As the difference falls well within the experimental uncertainties due to the usage of
different experimental methods in determining the reference rate coefficients by various groups,
they were averaged in our study. The average rate coefficient for the reaction R2 at 298K in the
present study is obtained to be $k_{R2-298K}^{Expt}$=(4.0±0.3)×10$^{-10}$ cm$^3$molecule$^{-1}$s$^{-1}$. The measured rate
coefficient is reasonably in good agreement with the reported one by Blanco et al. at 298 K,
(3.15±0.5)×10$^{-10}$ cm$^3$molecule$^{-1}$s$^{-1}$. Thus an averaged rate coefficient at all the temperatures was
used to fit the Arrhenius equation and is shown in the Figure 6 along with the available rate
coefficients for this reaction. The temperature dependent rate coefficient for the reaction R2 is
obtained to be $k_{R2-(298-363K)}^{Expt}$ = (4.6±2.4)×10$^{-11}$ exp[(646±171)/T] cm$^3$molecule$^{-1}$s$^{-1}$.
**3.2.2. Computational**

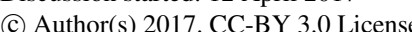

**Structure 2: 5-hexen-2-one**

The optimized geometries of the reactant, pre-reactive complexes, transition states and products
are shown in Figure 7. In 5-hexen-2-one, two addition channels namely TS1a (terminal carbon)





and TS2a (middle carbon) and abstraction channels TS1 to TS10 were found. As shown in the
structure 2, addition transition states are represented by suffix 'a' and the transition states for
hydrogen abstractions are labeled from TS1 to TS10 as identified above. The Cl atom addition
across the double bond via transition states TS1a and TS2a forms secondary and primary radicals
(P1a and P2a) respectively. Hydrogen abstraction reactions form products P1 to P10 via TSs TS1
to TS10 respectively. In case of hydrogen abstraction reactions, the leaving –C-H bond was
stretched up to 30 to 41% when compared with normal -C-H bond length in the substrate. The
stationary point energies were refined at CCSD(T)/6-31+G(d, p) using the optimized geometries
obtained at MP2/6-311++G(d, p) level of theory. Thus refined energies were used in dual level
CVT calculations to compute the rate coefficients for the reaction over the temperature range of
275-400K. The potential energy surface for this reaction is given in Figure 8. The contribution
from the addition channels via transition states TS1a and TS2a is expected to be maximum and
they are submerged channels. The corresponding pre-reactive complexes were also identified for
these submerged reaction channels and were used in the rate coefficient calculations. As
explained in the reaction R1, the global rate coefficient for reaction R2 was obtained by linear
summation of rate coefficients for the individual channels. The rate coefficient for reaction R2 at
298K was calculated to be $k_{R2-298K}^{Theory}$= 5.56×10$^{-10}$ cm$^3$molecule$^{-1}$s$^{-1}$. Theoretically calculated rate
coefficient is larger by 45% when compared with the reported one by Blanco et al. at 298K,
(3.15±0.5)×10$^{-10}$  cm$^3$molecule$^{-1}$s$^{-1}$. Also, it is larger than the experimentally measured one by
22%. May be this is due to the uncertainties associated with the submerged transition states and
theoretically calculated pre-exponential factors. Here it should be noted that the uncertainties in
the calculated energies of adducts and transition states can critically effect the calculated rate
coefficients. On the other hand, the pre-exponential factor which depends upon the partition
functions of reactants and transition states are estimated in the calculations. The computed rate
coefficients were used to fit the Arrhenius equation by linear least squares method and is given in
Figure 6 along with the experimentally measured ones. The theoretically obtained temperature
dependent  rate  coefficient  for  the  reaction  R2  is  $k_{R2-(275-400K)}^{Theory}$  =  (3.20±0.27)×10$^-$
$^{12}$exp[(1535±24)/T] cm$^3$ molecule$^{-1}$s$^{-1}$. The activation energy for the reaction R2 estimated by
theory is double than that of experimentally derived one. The pre-exponential factor estimated
theoretically is lower than the experimentally derived one by an order of magnitude.



**3.3. Reaction of Cl atoms with 3-penten-2-one**
Although no data is available on the concentration of 3-penten-2-one in the atmosphere, the
concentration may not be insignificant as the source is biomass. Therefore, measurement of the
kinetics of the oxidation is useful. However, the vapor pressure of this compound is very less
(less than a Torr at 298K). Therefore, experiments were not performed for this molecule. To
understand the mechanism of oxidation of series of unsaturated ketones, 3-penten-2-one was also
studied using computational methods along with 4-hexen-3-one and 5-hexen-2-one.

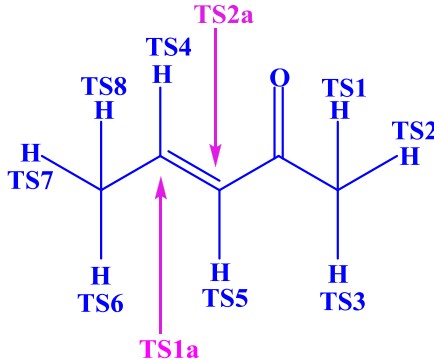

**Structure 3: 3-penten-2-one**

The optimized (MP2/6-311++G(d, p)) geometries of reactant (3-penten-2-one), pre-reactive
complexes, transition states and products are shown in Figure 9. In 3-penten-2-one, two addition
transition states namely TS1a and TS2a and eight hydrogen abstraction channels namely TS1 to
TS8 were identified and are shown in the structure 3.  As shown in potential energy level
diagram Figure 10, Cl atom addition at TS1a and TS2a leads to the formation of secondary
radicals P1a and P2a respectively. The transition states for the addition channels are submerged
and the corresponding pre-reactive complexes were optimized at the same level of theory. The
reaction is dominated by the addition pathways. During hydrogen abstraction reactions, the
leaving –C-H bond was elongated from 16% to 34% when compared with normal -C-H bond in
the reactant. The energies of all the stationary points were refined at CCSD(T)/6-31+G(d, p)
using the optimized geometries obtained at MP2/6-311++G(d, p) level of theory. The refined
energies were used in dual level CVT calculations to compute the rate coefficients for the
reaction R3 over the temperature range of 275-400K. The global rate coefficient was obtained by





summing the rate coefficients of individual pathways. The rate coefficient of the reaction R3 at
298 K was obtained to be $k_{R3-298K}^{Theory}$= 2.40×10$^{-10}$ cm$^3$molecule$^{-1}$s$^{-1}$. Theoretically calculated rate
coefficient shows the very good agreement with the reported one by Blanco et al. at 298K,
(2.53±0.54)×10$^{-10}$ cm$^3$molecule$^{-1}$s$^{-1}$. The calculated global rate coefficients were used to get the
temperature dependence of the reaction R3 and derived Arrhenius expression is $k_{R3-(275-400K)}^{Theory}$ =
(2.5±0.2)×10$^{-12}$exp [(1363±26)/T] cm$^3$molecule$^{-1}$s$^{-1}$. Arrhenius plot is shown in Figure 11 which
shows negative temperature dependence over the studied temperature range.
**3.4. Addition vs abstraction**
As mentioned in the discussions of each reactions, the addition pathways contribute more
towards the global rate, as all the corresponding transition states are submerged ones. The
observed negative temperature dependence for the studied reactions R1, R2 and R3 is due to the
stabilized submerged transition states. In a review on the reactions of unsaturated molecules with
Cl atoms, (Taatjes, 1999) concluded that negative temperature dependence for the addition of Cl
atoms across the double bond to be the most favourable pathway. In present investigation, for the
reactions of Cl atoms with 4-hexen-3-one, 5-hexen-2-one and 3-penten-2-one have shown
negative temperature dependence which is in consistent with the findings of Taatjes, 1999.
The contributions of the abstraction channels are negligible when compared to the addition
channels. The temperature dependence of branching ratios of all the three reactions are given in
Figures 12, 13 and 14. It is obvious from these Figures that both the addition channels across the
double bond contribute maximum to the reaction. An abstraction channels seem to have a
negligible contribution to the reaction in the studied temperature range. However, it should be
noted that, the contribution of abstraction seems to be increasing with the temperature. The
trends of the addition rate coefficients in case of the compounds containing double bonds
adjacent to the C=O group (4-hexen-3-one and 3-pentene-2-one) were observed to follow similar
trends. The rate coefficients for the addition of Cl atom at α position to C=O bond in case of both
the molecules via transition state TS2a were observed to be increasing with the temperature.
However, the rate coefficients for the addition at β position to C=O bond were observed to be
decreasing with the increase in temperature. In both these molecules, β carbon is connected to
CH$_3$ group, which is an electron donating group. The C-C bond rotation will be more favorable



at higher temperatures, which would lead to elongation of the bond and therefore, the influence
of the electron donating group reduces with the increase in temperature. Consequently, the
transition state for the addition at β position would become "tight". In other words, the activation
barrier height would increase and therefore, rate coefficient would decrease. In case of 5-hexen-
2-one, due to the additional two $CH_2$ groups between double bond and C=O group, and the
absence of $CH_3$ group at the terminal carbon, the above said influence will not play any role.
And therefore, it is expected that, the addition across the double bond via transition states TS1a
and TS2a should go with almost equal rates. This can be seen in the branching ratio plot for
reaction R2.
**3.5. Reactivity trend**
The rate coefficients for the reactions of Cl atoms, OH and $NO_3$ radicals (Atkinson et al., 2003;
Smith et al., 1996; Holloway et al., 2005; Neeb et al., 1998 and Canosa-Mas et al., 2005) with
analogous molecules are given in Table 9. From Table 9, it is very clear that for the reactions of
Cl atoms with unsaturated compounds, from propene to 1-pentene there is an increment in the
rate coefficient due to insertion of two -$CH_2$- groups. A significant drop in the rate coefficients
for the reaction of Cl atoms, OH and $NO_3$ radicals was observed with insertion of –C=O group
into the alkene. By insertion of -$CH_2$ group, the rate coefficient is increased from 3-buten-2-one
to 3-penten-2-one by factor of 1.5 and from 3-penten-2-one to 4-hexen-3-one by factor of 1.2
which indicates that the reactivity increases by insertion of electron donating group (-$C_2H_4$-)
whereas it decreases with insertion of a strong deactivating group (-C=O). The same trend can be
observed for the reaction with $NO_3$ radicals that is insertion of -$C_2H_4$- group into propylene to
give 1-pentene increases the reactivity by a factor of 1.6 and insertion of C=O group into 1-
pentene to give 5-hexen-2-one decreases the reactivity by a factor of 1.5. The similar effects are
observed for the reactions of these compounds with OH radicals.
In case of OH radical reactions, –$C_2H_4$ group is inserted into 3-buten-2-one to give 5-hexen-2-
one which increases the reactivity of 5-hexen-2-one. The α-substitution with two methyl groups
in 5-hexen-2-one to give 6-methyl-5-hepten-2-one increases the reactivity of 5-hexene-2-one
which indicates that the preferential electrophilic addition mechanism for OH reactions.
Structure Activity Relations (SAR) (Kwok et al., 1995 and Mellouki et al., 2003) also supports
this mechanism and according to SAR the reaction of Cl atoms and OH radicals with these





unsaturated ketones proceeds via addition across the double bond as a predominant channel and
only 15% accounts for hydrogen abstraction.
4-hexen-3-one ($C_6H_{10}$) and 5-hexen-2-one ($C_6H_{10}$) are structural isomers but the experimentally
measured rate coefficient in present investigation for the reaction of Cl atoms with 4-hexen-3-
one ($(5.54\pm0.41)\times10^{-10}cm^3molecule^{-1}s^{-1}$) is higher than the reaction with 5-hexen-2-one
($(4.0\pm0.3)\times10^{-10}cm^3molecule^{-1}s^{-1}$). The same trend was observed by Blanco et al. in the case of
OH radical reactions. The rate coefficient for the reaction of 4-hexene-3-one with OH ($9.04\times10^{-11}$
$cm^3molecule^{-1}s^{-1}$) is higher than the rate coefficient for the reaction with 5-hexene-2-one
($5.18\times10^{-11}$ $cm^3molecule^{-1}s^{-1}$). Moreover, the double bond in 4-hexen-3-one is in conjugation
with carbonyl group and upon Cl atom addition across double bond forms more stable radical
which makes 4-hexen-3-one to be more reactive than the 5-hexen-2-one.
**3.6. Feasibility of a reaction**
To understand the feasibility and spontaneity of the reactions R1, R2 and R3 in terms of
thermodynamic parameters, standard Gibb's free energies, enthalpies and entropies were
calculated at MP2/6-311++G (d, p) level of theory and are given in Tables 3, 5 and 8. From these
Tables, it is very clear that the Cl atom addition reactions are more favorable than hydrogen
abstraction reactions which are in consistence with our findings using kinetic parameters (cf
section 3.4). In reaction R1 (4-hexen-3-one + Cl), addition through TS1a and TS2a are equally
spontaneous and exothermic. In case of reaction R2 (5-hexen-2-one + Cl) also, the addition of Cl
atom through TS1a and TS2a are equally spontaneous and exothermic. The same trend was
observed in case of reaction R3 (3-penten-2-one + Cl) as well.
As far as the abstraction of hydrogen channels are considered, all of them were found to be
endothermic. Among all the possible hydrogen abstraction channels, abstraction of hydrogens
attached to olefinic carbon are more endothermic. From Tables 3, 5 and 8, it is clear that there is
a sudden jump in the $\Delta H^0$ values for these channels (given with BOLD fonts, TS6 and TS7 in
R1; TS8, TS9 and TS10 in R2; and TS4 and TS5 in case of R3 reactions).



**3.7. Degradation mechanism**
The reactions of unsaturated compounds with Cl atoms mainly proceeds via initial addition to the
C=C bond (s) which is analogous to the other atmospheric oxidants such as OH, $NO_3$ and $O_3$
(Atkinson et al., 2003). In the product analysis of OH initiated reaction with double bonded
molecule such as isoprene, Atkinson et al., 2003 has observed addition products like methyl
vinyl ketone, HCHO, methacrolein, 3-methylfuran and C5-hydroxycarbonyls; and abstraction
products like 1,4-hydroxyaldehyes with loss of water at room temperature and atmospheric
pressure. In case of $NO_3$ radicals (Canosa-Mas et al., 2005), they undergo addition reactions and
at low pressure, they form chemically activated nitrooxyalkyl radical and can decompose to an
oxirane and $NO_2$ in competition with collisional stabilization. Similarly, reaction of $O_3$ with
unsaturated molecules proceed via initial addition which forms primary ozonide, which further
decompose to form Criegee intermediates (Neeb et al., 1998) via two different pathways.
The oxidation mechanism is proposed for one test molecule (5-hexen-2-one) based on the above
observations and is given in Figure 15. The reactions of Cl atoms with test molecules occur
primarily via addition across double bond to form chloroalkyl radicals. When these chloroalkyl
radicals reacts with $O_2$ yields chloroalkyl peroxy radicals and in presence of $NO_x$, these
chloroalkyl peroxy radicals form chloroalkoxy radicals. Finally, they form carbonyl compounds
as end products.
**3.8. Atmospheric implications**
These unsaturated ketones would be degraded in the atmosphere due to their reactions with
oxidants such as OH radicals, Cl atoms, $NO_3$ radicals and $O_3$ molecules. Therefore, cumulative
atmospheric life times of 4-hexen-3-one, 5-hexen-2-one and 3-pentene-2-one with respect to OH
radicals, Cl atoms, $NO_3$ radicals and $O_3$ molecules were calculated using the following equation

$$\frac{1}{\tau_{eff}} = \frac{1}{\tau_{OH}} + \frac{1}{\tau_{Cl}} + \frac{1}{\tau_{O_3}} + \frac{1}{\tau_{NO_3}}$$

where $\tau_{eff}$ is the cumulative lifetime of the chemical species, $\tau_{OH}$, $\tau_{Cl}$, $\tau_{O3}$ and $\tau_{NO3}$ are rate
coefficients for the reactions of test molecules with OH radicals, Cl atoms, $O_3$ and $NO_3$ radicals
respectively. The global atmospheric concentrations used in present investigation are $2\times10^6$





radical cm$^{-3}$, 5×10$^8$ radical cm$^{-3}$ and 7×10$^{11}$ molecules cm$^{-3}$ for OH radical, NO$_3$ radicals and
ozone respectively (Atkinson, 2000) and depicted in Table 10. To know the importance of the Cl
atom reactions, the atmospheric lifetimes of the test molecules were estimated with respect to
their reactions with Cl atoms both in ambient conditions (1.00×10$^3$ molecules cm$^{-3}$) (Singh et al.,
1996) and marine boundary layer (1.30×10$^5$ molecules cm$^{-3}$) (Spicer et al., 1998). In ambient
conditions the lifetimes of 4-hexen-3-one, 5-hexen-2-one and 3-pentene-2-one are 8, 12 and 19
days respectively. In the marine boundary layer the lifetimes of 4-hexen-3-one, 5-hexen-2-one
and 3-pentene-2-one are estimated as 3.5, 5.3 and 8.7 hours respectively. Since the Cl atom
reactions are important mainly in marine boundary layer and polluted urban areas where the Cl
atom concentration reaches 1.3×10$^5$ atoms cm$^{-3}$ was considered when compared with other
oxidants in the Table 10. From Table 10, it is very clear that the reactions of Cl atoms with test
molecules are much faster (in the order of 10$^{-10}$ cm$^3$ molecule$^{-1}$ s$^{-1}$) when compared with the
reactions of OH radicals (in the order of 10$^{-11}$ cm$^3$ molecule$^{-1}$ s$^{-1}$) and NO$_3$ radicals (in the order
of 10$^{-14}$ cm$^3$ molecule$^{-1}$ s$^{-1}$) and O$_3$ molecules (in the order of 10$^{-17}$ cm$^3$ molecule$^{-1}$ s$^{-1}$). However,
the atmospheric degradation of these ketones due to their reactions with Cl atoms are less when
compared with OH radicals. This is because of lower concentrations of Cl atoms when compared
with that of OH radicals in the Earth's atmosphere. However, the cumulative lifetimes of these
test molecules are few hours. As these molecules are short lived they would not contribute to
global warming in any time horizons.
**4. Conclusions**
In this study, reactions of Cl atoms with 4-hexen-3-one, 5-hexen-2-one and 3-penten-2-one were
investigated (Reactions labeled R1, R2 and R3 respectively). R1 and R2 were investigated
experimentally in the temperature range of 298-363K whereas R1, R2 and R3 were studied
computationally. The rate coefficients for all the reactions were calculated by CVT/SCT coupled
with CCSD(T)/6-31+G(d,p)//MP2/6-311++G(d,p) basis sets. Addition of Cl atom across the
double bond in all the reactions predominates and abstraction of hydrogen would have very
insignificant contribution to the overall reaction. From both experimental and theoretical
measurements, negative temperature dependence was observed over the studied temperature
range because of the submerged transition states for addition channels. Thermodynamically these
addition reactions are more feasible and spontaneous. The cumulative lifetimes of the test


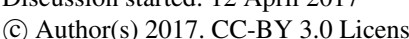


molecules are very low and they are lost within few hours as soon as they are released into the
atmosphere.
**Supporting information:**
The optimized parameters for the reactants, pre-reactive complexes, transition states and
products are given in Tables S-1-1 to S-5-22 and the vibrational frequencies are given in Tables
S-2-1 to S-6-4 for the reactions of Cl atoms with 4-hexen-3-one, 5-hexen-2-one and 3-penten-2-
one.
**Acknowledgement**
The authors cordially thank the Department of Science and Technology (DST), Government of
India, for the financial support. We also thank Mr.V.Ravichandran, High Performance
Computing Environment (HPCE), IITMadras for the help in providing the computational
resources to carry out the calculations.

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

**Table 1**: Relative rate measurements for the reaction of Cl atoms with 4-hexen-3-one over the
temperature range of 298-363K at 760 Torr in $N_2$ relative to 1-pentene, 1,3-butadiene and
isoprene.

| Temperature (K) | Reference compound | $(k_{sample}/k_{reference}) \pm 2\sigma$ | $(k_{sample}/k_{reference})_{Average} \pm 2\sigma$ | $(k \pm 2\sigma) \times 10^{-10} \pm 2\sigma$ cm$^3$molecule$^{-1}$s$^{-1}$ | $(k \pm 2\sigma) \times 10^{-10} \pm 2\sigma$ cm$^3$molecule$^{-1}$s$^{-1}$ | Literature $(k \pm 2\sigma) \times 10^{-10} \pm 2\sigma$ (cm$^3$molecule$^{-1}$s$^{-1}$) at 298K |
|---|---|---|---|---|---|---|
| 298±2 | 1,3-butadiene | 1.55±0.21 | 1.46±0.36 | 5.08±0.36 | 5.54±0.41 | 3.00±0.58 [a] |
|  |  | 1.43±0.24 |  |  |  |  |
|  |  | 1.64±0.27 |  |  |  |  |
|  |  | 1.22±0.11 |  |  |  |  |
|  | isoprene | 1.63±0.15 | 1.61±0.03 | 5.84±0.03 |  |  |
|  |  | 1.62±0.11 |  |  |  |  |
|  |  | 1.60±0.13 |  |  |  |  |
|  | 1-pentene | 1.12±0.12 | 1.21±0.26 | 5.71±0.26 |  |  |
|  |  | 1.09±0.13 |  |  |  |  |
|  |  | 1.33±0.10 |  |  |  |  |
|  |  | 1.33±0.10 |  |  |  |  |





| | | | | | |
|---|---|---|---|---|---|
| 310±2 | 1-pentene | 0.93±0.01 | 0.89±0.09 | 3.82±0.09 | 3.97±0.21 |
| | | 0.86±0.06 | | | |
| | isoprene | 1.24±0.10 | 1.29±0.07 | 4.13±0.07 | |
| | | 1.37±0.10 | | | |
| | | 1.26±0.09 | | | |
| 330±2 | 1-pentene | 0.79±0.04 | 0.80±0.02 | 2.95±0.02 | 3.15±0.28 |
| | | 0.81±0.04 | | | |
| | isoprene | 1.24±0.10 | 1.15±0.07 | 3.36±0.07 | |
| | | 1.13±0.09 | | | |
| | | 1.09±0.08 | | | |
| 350±2 | 1-pentene | 0.74±0.05 | 0.74±0.01 | 2.42±0.01 | 2. 63±0.29 |
| | | 0.75±0.06 | | | |
| | isoprene | 0.98±0.07 | 1.06±0.10 | 2.84±0.10 | |
| | | 1.19±0.10 | | | |
| | | 1.01±0.08 | | | |
| 363±2 | isoprene | 0.87±0.4 | 0.86±0.03 | 3.01±0.03 | 2.6±0.58 |
| | | 0.88±0.2 | | | |
| | | 0.85±0.6 | | | |
| | 1-pentene | 0.74±0.02 | 0.72±0.03 | 2.18±0.03 | |
| | | 0.72±0.11 | | | |
| | | 0.71±0.10 | | | |

2    [a]Blanco et al.





**Table 2**: Calculated total CVT/SCT rate coefficients (cm$^3$ molecule$^{-1}$ s$^{-1}$) for the reaction of Cl
atoms with 4-hexen-3-one obtained at the CCSD(T)/6-31+G(d,p)//MP2/6-311++G(d,p) level of
theory.

| Temperature (K) | k (cm$^3$ molecule$^{-1}$ s$^{-1}$) |
|---|---|
| 275 | $5.81\times10^{-10}$ |
| 298 | $3.66\times10^{-10}$ |
| 325 | $2.33\times10^{-10}$ |
| 350 | $1.64\times10^{-10}$ |
| 375 | $1.22\times10^{-10}$ |
| 400 | $9.49\times10^{-11}$ |

**Table 3**: Barrier heights [$\Delta E^{0\ddagger}$, kcal mol$^{-1}$], kcal mol$^{-1}$ at 298K], Gibbs free energy [$\Delta G^0$ (298K),
kcal mol$^{-1}$], heat of reaction [$\Delta H^0$ (298K) and entropy of reaction [$\Delta S^0$ (298K), cal mol$^{-1}$ K$^{-1}$] for
the reaction of Cl atoms with 4-hexen-3-one at the MP2/6-311++G(d, p) level of theory.

| TSs | $\Delta E^{0\ddagger}$,kcal mol$^{-1}$ | $\Delta H^0$, kcal mol$^{-1}$ | $\Delta G^0$, kcal mol$^{-1}$ | $\Delta S^0$calmol$^{-1}$ K$^{-1}$ |
|---|---|---|---|---|
| TS1a | -3.80 | -15.85 | -7.81 | -26.96 |
| TS2a | -3.45 | -16.35 | -7.42 | -29.94 |
| TS1 | 10.63 | 2.21 | -0.34 | 8.58 |
| TS2 | 10.63 | 2.21 | -0.34 | 8.58 |
| TS3 | 11.36 | 2.21 | -0.34 | 8.58 |
| TS4 | 10.93 | 4.01 | 1.57 | 8.17 |
| TS5 | 10.93 | 4.01 | 1.57 | 8.17 |
| **TS6** | **17.97** | **15.88** | **13.82** | 6.91 |
| **TS7** | **17.97** | **21.86** | **19.73** | 7.12 |
| TS8 | 9.68 | -3.04 | -5.08 | 6.85 |
| TS9 | 9.68 | -3.04 | -5.08 | 6.85 |
| TS10 | 9.68 | 4.40 | 2.40 | 6.71 |



1 **Table 4**: Relative rate measurements for the reaction of Cl atoms with 5-hexen-2-one over the

2 temperature range of 298-363K at 760 Torr in $N_2$ with reference to isoprene and 1-pentene.

| $T$ (K) | Reference compound | $(k_{sample}/k_{reference})\pm2\sigma$ | $(k_{sample}/k_{reference})_{Average}\pm2\sigma$ | $(k\pm2\sigma)\times10^{-10}$ ($cm^3molecule^{-1}s^{-1}$) | $(k_{Average}\pm2\sigma)\times10^{-10}$ ($cm^3molecule^{-1}s^{-1}$) | Lite.$k\times10^{-10}$ ($cm^3molecule^{-1}s^{-1}$) at 298K |
|---|---|---|---|---|---|---|
| 298±2 | 1-pentene | 0.87±0.09 | 0.91±0.08 | 4.27±0.08 | 4.00±0.37 | 3.15±0.5 [a] |
| | | 0.95±0.08 | | | | |
| | | 0.91±0.08 | | | | |
| | isoprene | 1.18±0.09 | 1.09±0.11 | 3.74±0.11 | | |
| | | 1.14±0.1 | | | | |
| | | 0.97±0.16 | | | | |
| 310±2 | isoprene | 1.06±0.08 | 1.09±0.09 | 3.51±0.09 | 3.77±0.35 | |
| | | 1.13±0.06 | | | | |
| | 1-pentene | 0.94±0.08 | 0.94±0.08 | 4.04±0.08 | | |
| | | 0.95±0.09 | | | | |
| 330±2 | isoprene | 1.05±0.06 | 1.07±0.05 | 3.11±0.05 | 3.11±0.01 | |
| | | 1.09±0.08 | | | | |
| | 1-pentene | 0.84±0.06 | 0.84±0.01 | 3.12±0.01 | | |
| | | 0.85±0.09 | | | | |
| 350±2 | isoprene | 1.17±0.16 | 1.14±0.07 | 3.07±0.07 | 3.00±0.55 | |
| | | 1.12±0.13 | | | | |
| | 1-pentene | 0.92±0.10 | 0.91±0.02 | 2.94±0.02 | | |
| | | 0.89±0.09 | | | | |
| 363±2 | isoprene | 0.98±0.07 | 1.05±0.11 | 2.67±0.11 | 2.66±0.03 | |
| | | 0.99±0.09 | | | | |
| | | 1.18±0.08 | | | | |
| | 1-pentene | 0.89±0.05 | 0.88±0.02 | 2.65±0.02 | | |



| | | 0.88±0.07 | | | |
| | | 0.87±0.06 | | | |

[a]Blanco et al.
**Table 5**: Barrier heights [$\Delta E^{0\ddagger}$, kcal mol$^{-1}$ at 298K], heat of reaction [$\Delta H^0$ (298K), kcal mol$^{-1}$],
Gibbs free energy [$\Delta G^0$ (298K), kcal mol$^{-1}$] and entropy of reaction [$\Delta S^0$ (298K), cal mol$^{-1}$ K$^{-1}$]
for the reaction of Cl atoms with 5-hexen-2-one obtained at the MP2/6-311++G(d, p) level of
theory.

| TSs | $\Delta E^{0\ddagger}$, kcal mol$^{-1}$ | $\Delta H^0$, kcal mol$^{-1}$ | $\Delta G^0$, kcal mol$^{-1}$ | $\Delta S^0$calmol$^{-1}$ K$^{-1}$ |
|---|---|---|---|---|
| Ts1a | -3.63 | -14.25 | -6.39 | -26.33 |
| Ts2a | -3.68 | -15.02 | -6.34 | -29.09 |
| Ts1 | 15.83 | 6.42 | 5.13 | 5.81 |
| Ts2 | 18.13 | 6.95 | 5.38 | 5.25 |
| Ts3 | 14.49 | 6.28 | 4.95 | 5.08 |
| Ts4 | 10.16 | 4.85 | 2.59 | 3.25 |
| Ts5 | 9.56 | 4.26 | 2.47 | 3.86 |
| Ts6 | 9.82 | 5.13 | 6.87 | 7.15 |
| Ts7 | 7.30 | 4.95 | 6.51 | 5.25 |
| **Ts8** | **15.8** | **11.56** | **8.56** | 6.19 |
| **Ts9** | **18.06** | **12.45** | **9.47** | 7.48 |
| **Ts10** | **20.42** | **12.35** | **9.66** | 5.66 |





**Table 6**: Calculated total CVT/SCT rate coefficients (cm$^3$ molecule$^{-1}$ s$^{-1}$) for the reaction of Cl
atoms with 5-hexen-2-one obtained at CCSD(T)/6-31+G(d,p)//MP2/6-311++G (d, p) level of
theory.

| Temperature(K) | k (cm$^3$ molecule$^{-1}$ s$^{-1}$) |
|---|---|
| 275 | $1.09 \times 10^{-09}$ |
| 298 | $5.56 \times 10^{-10}$ |
| 325 | $3.85 \times 10^{-10}$ |
| 350 | $2.44 \times 10^{-10}$ |
| 375 | $1.57 \times 10^{-10}$ |
| 400 | $1.00 \times 10^{-10}$ |

**Table 7**: Calculated total CVT/SCT rate coefficients (cm$^3$ molecule$^{-1}$ s$^{-1}$) for the reaction of Cl
atoms with 3-penten-2-one obtained at the CCSD(T)/6-31+G(d, p)//MP2/6-311++G (d, p) level
of theory.

| Temperature(K) | k (cm$^3$ molecule$^{-1}$ s$^{-1}$) |
|---|---|
| 275 | $3.51 \times 10^{-10}$ |
| 298 | $2.40 \times 10^{-10}$ |
| 325 | $1.66 \times 10^{-10}$ |
| 350 | $1.26 \times 10^{-10}$ |
| 375 | $9.89 \times 10^{-11}$ |
| 400 | $8.07 \times 10^{-11}$ |



1    **Table 8**: Barrier heights [$\Delta E^{0\ddagger}$, kcal mol$^{-1}$ at 298K], heat of reaction [$\Delta H^0$ (298K), kcal mol$^{-1}$], Gibbs

2    free energy [$\Delta G^0$ (298K), kcal mol$^{-1}$] and entropy of reaction [$\Delta S^0$ (298K), cal mol$^{-1}$ K$^{-1}$] for the reaction

3    of Cl atoms with 3-penten-2-one at the MP2/6-311++G (d, p) level of theory.

| TSs | $\Delta E^{0\ddagger}$,kcal mol$^{-1}$ | $\Delta H^0$, kcal mol$^{-1}$ | $\Delta G^0$, kcal mol$^{-1}$ | $\Delta S^0$calmol$^{-1}$K$^{-1}$ |
|---|---|---|---|---|
| Ts1a | -4.12 | -15.23 | -7.57 | -25.70 |
| Ts2a | -3.75 | -15.57 | -6.97 | -28.85 |
| Ts1 | 13.94 | 4.32 | 3.13 | 6.81 |
| Ts2 | 14.23 | 4.15 | 3.09 | 6.24 |
| Ts3 | 14.23 | 4.15 | 3.09 | 6.24 |
| **Ts4** | **15.85** | **14.89** | **11.57** | 8.23 |
| **Ts5** | **19.45** | **14.25** | **11.62** | 8.48 |
| Ts6 | 9.82 | 2.38 | 4.25 | 5.87 |
| Ts7 | 10.17 | 2.61 | 4.60 | 6.69 |
| Ts8 | 10.17 | 2.61 | 4.60 | 6.69 |



1    **Table 9:** Reactivity of series of unsaturated ketones with Cl atoms, OH and NO$_3$ radicals at

2    298K.

| Molecule | $k_{Cl}$ (cm$^3$ molecule$^{-1}$ s$^{-1}$) | $k_{OH}$ (cm$^3$ molecule$^{-1}$ s$^{-1}$) | $k_{NO3}$ (cm$^3$ molecule$^{-1}$ s$^{-1}$) |
|---|---|---|---|
| propene | $2.7\times10^{-10}$ [a] | $2.63\times10^{-11}$ [d] | $9.5\times10^{-15}$ [d] |
| 1-pentene | $4.0\times10^{-10}$ [a] | $3.14\times10^{-11}$ [d] | $1.51\times10^{-14}$ [d] |
| 3-buten-2-one | $0.99\times10^{-10}$ [b] | $1.86\times10^{-11}$ [d] | $5.47\times10^{-16}$ [d] |
| 3-penten-2-one | $2.40\times10^{-10}$ [c] | $7.22\times10^{-11}$ [e] | $1.03\times10^{-14}$ [e] |
| 4-hexen-3-one | $3.66\times10^{-10}$ [c] | $9.04\times10^{-11}$ [e] | $0.63\times10^{-14}$ [e] |
| 5-hexen-2-one | $5.56\times10^{-10}$ [c] | $5.18\times10^{-11}$ [e] | $2.16\times10^{-14}$ [e] |
| 6-methyl-5-hepten-2-one | - | $1.57\times10^{-10}$ [a] | $7.5\times10^{-12}$ [a] |

4    [a] Smith et al., [b] Canosa-Mas et al., [c] Present work, [d] Atkinson et al., [e] Wang et al.





**Table 10**: Atmospheric lifetimes (τ) calculated for series of unsaturated ketones with different
atmospheric oxidizing agents at 298K.

| Molecule | Cl atoms[a] | | OH radicals [b] | | NO₃ radicals [c] | | O₃ radicals[c] | | Cumulative life times |
|---|---|---|---|---|---|---|---|---|---|
| | $k_{Cl} \times 10^{10}$ | τ (hr) | $k_{OH} \times 10^{11}$ | τ (hr) | $k_{NO3} \times 10^{14}$ | τ (hr) | $k_{O3} \times 10^{17}$ | τ (hr) | $\tau_{eff}$ (hr) |
| **4-hexen-3-one** | 3.66 | 3.5 | 9.04 | 2 | 0.63 | 87 | 6.37 | 6 | 1.04 |
| **5-hexen-2-one** | 5.56 | 5.3 | 5.18 | 3 | 2.16 | 26 | 0.91 | 43 | 1.75 |
| **3-penten-2-one** | 2.40 | 8.7 | 7.22 | 2 | 1.03 | 54 | 2.95 | 13 | 1.41 |

[a]present work, [b]Blanco et al., [c]Wang et al.
**Table 11:** Branching ratios for the reactions of Cl atoms with 4-hexen-3-one, 5-hexen-2-one and
3-penten-2-one.

| T (K) | 4-hexen-3one | | | 5-hexen-2-one | | | 3-penten-2-one | | |
|---|---|---|---|---|---|---|---|---|---|
| | TS1a | TS2a | Total abstraction | TS1a | TS2a | Total abstraction | TS1a | TS2a | Total abstraction |
| 200 | 68.70 | 31.21 | 0.10 | 55.70 | 44.27 | 0.02 | 66.37 | 33.62 | 0.01 |
| 225 | 66.65 | 33.12 | 0.24 | 56.31 | 43.62 | 0.07 | 64.62 | 35.36 | 0.02 |
| 250 | 64.85 | 34.68 | 0.48 | 56.71 | 43.13 | 0.16 | 63.18 | 36.76 | 0.06 |
| 275 | 63.20 | 35.97 | 0.83 | 56.93 | 42.77 | 0.30 | 61.96 | 37.91 | 0.13 |
| 298 | 61.92 | 36.99 | 1.10 | 56.97 | 42.52 | 0.52 | 60.96 | 38.81 | 0.23 |
| 325 | 60.64 | 38.01 | 1.35 | 56.91 | 42.30 | 0.79 | 59.92 | 39.71 | 0.38 |
| 350 | 59.50 | 38.83 | 1.67 | 56.73 | 42.15 | 1.12 | 59.01 | 40.42 | 0.56 |
| 375 | 58.52 | 39.56 | 1.92 | 56.46 | 42.04 | 1.49 | 58.16 | 41.05 | 0.79 |
| 400 | 57.71 | 40.20 | 2.09 | 56.13 | 41.97 | 1.90 | 57.35 | 41.61 | 1.04 |



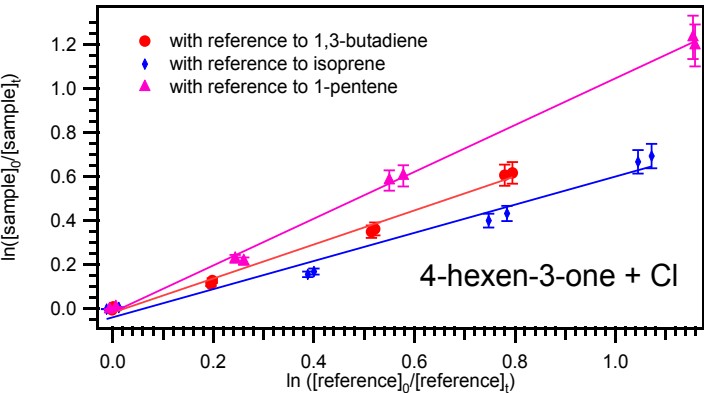

**Figure 1:** Plot of the relative decrease in the concentration of 4-hexen-3-one due to its reaction

with Cl atoms relative to 1-pentene, 1,3-butadiene and isoprene at 298K.

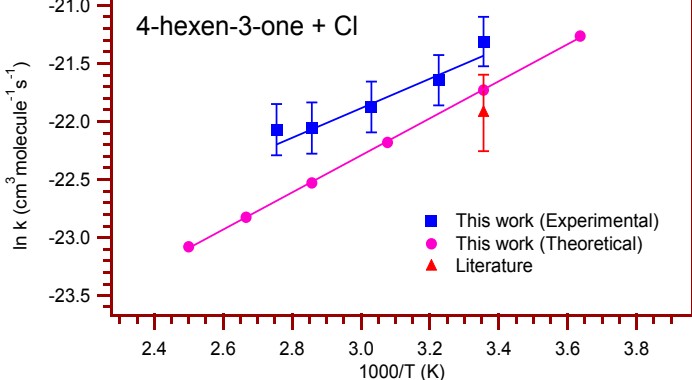

**Figure 2:** Arrhenius plot of CVT/SCT rate coefficients calculated at the CCSD(T)/6-31+G(d,

p)//MP2/6-311++G(d,p) level of theory between the temperatures 275 and 400 K and

experimentally measured rate coefficients between the temperatures of 298 and 363 K for the

reaction of Cl atoms with 4-hexen-3-one.







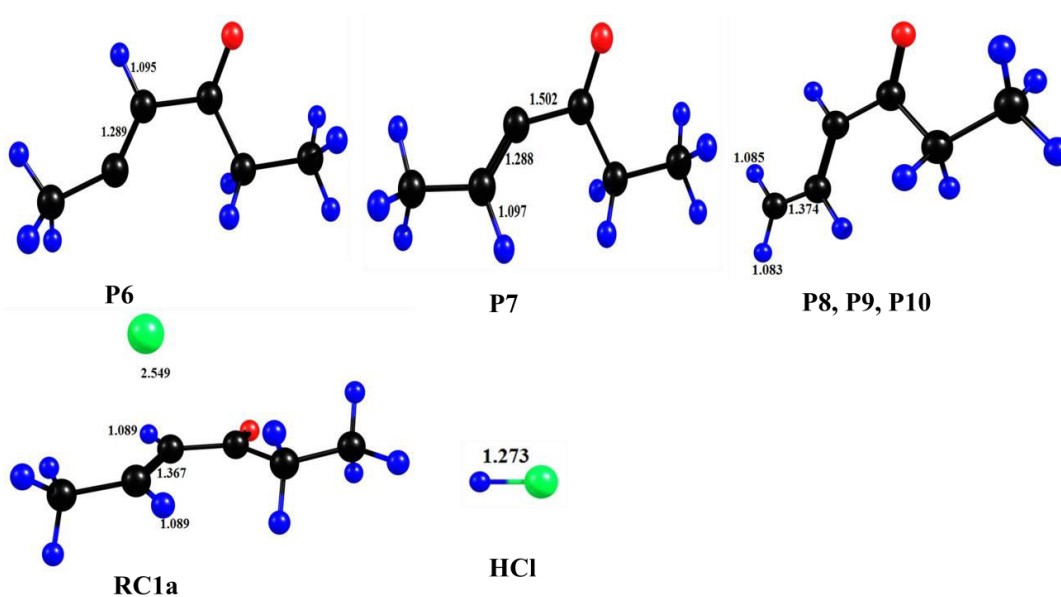

**Figure 3:** Optimized geometries of the reactants, pre-reactive complexes, transition states and products for the reaction of Cl atoms with 4-hexen-3-one obtained at MP2/6-311++G(d,p) level of theory. Black color represents carbon atoms, blue color represents hydrogen atoms, red color represents oxygen atoms and green color represents Cl atoms.



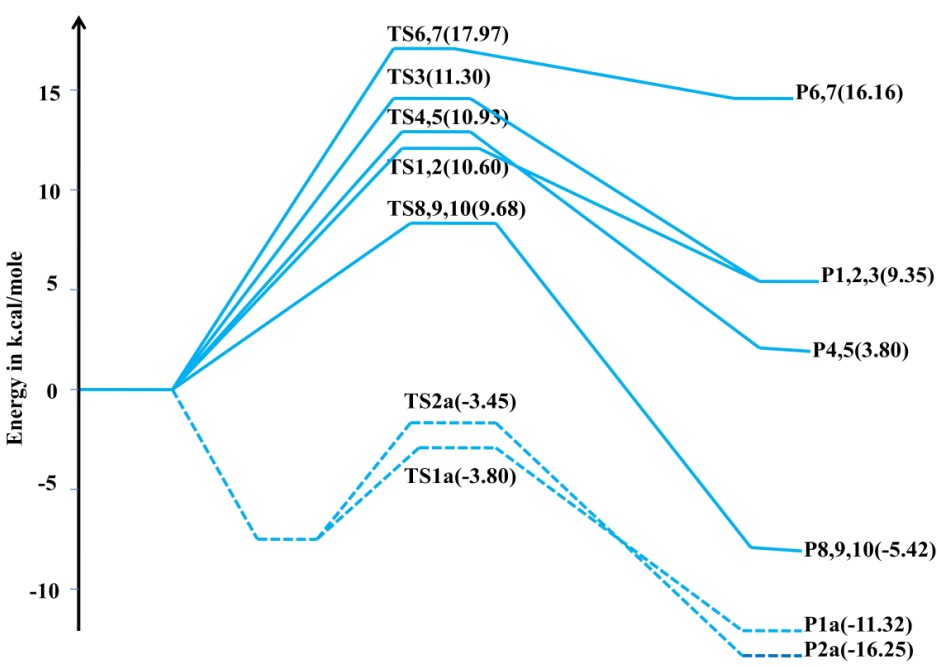

Figure 4: Potential energy diagram for the reaction of Cl atoms with 4-hexen-3-one obtained at

3    CCSD(T)/6-31+G(d,p)//MP2/6-311++G(d,p) level of theory at 298K.

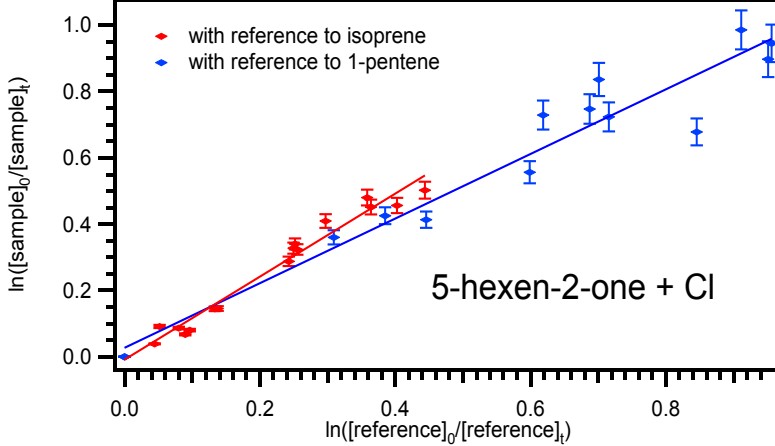

5    Figure 5: Plot of the relative decrease in the concentration of 5-hexen-2-one due to its reaction

6    with Cl atoms relative to isoprene and 1-pentene at 298K.



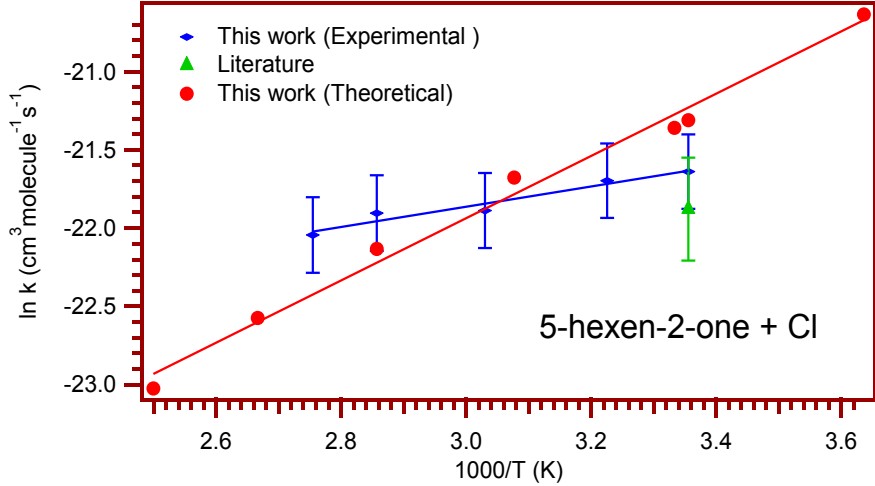

**Figure 6**: Arrhenius plot of CVT/SCT rate coefficients obtained at the CCSD(T)/6-31+G(d, p)//MP2/6-311++G(d,p) level of theory between the temperatures 275 and 400 K and experimentally measured rate coefficients between the temperatures of 298 and 363 K for the reaction of Cl atoms with 5-hexen-2-one.

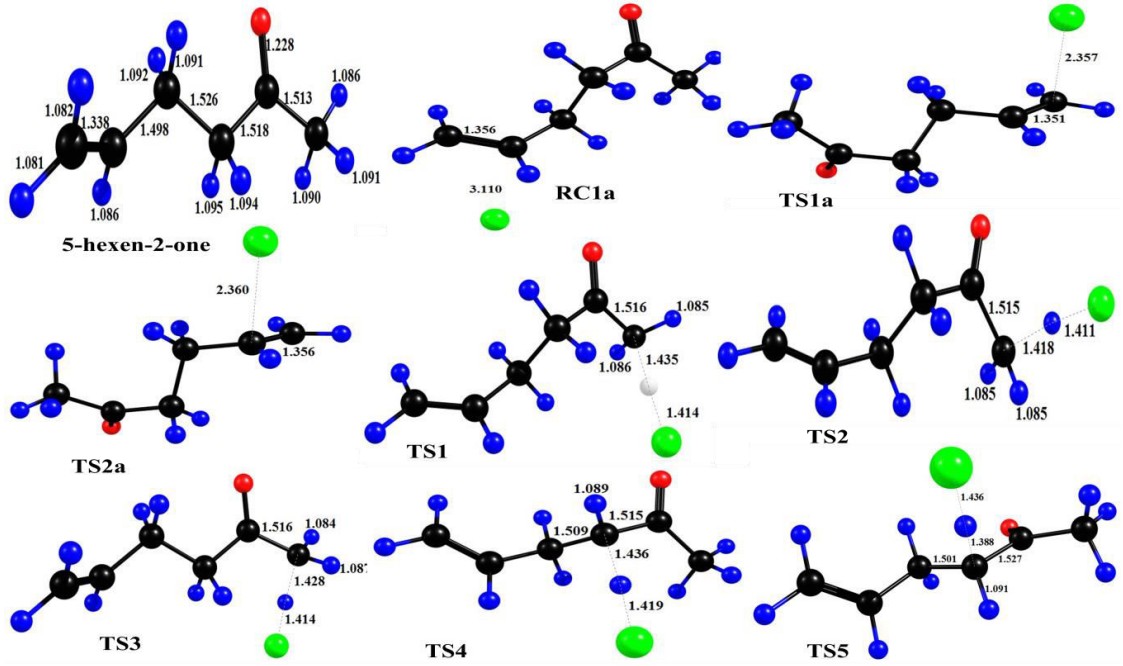



**Figure 7:** Optimized geometries of the reactants, pre-reactive complexes, transition states and
products for the reaction of Cl atoms with 5-hexen-2-one obtained at MP2/6-311++G(d, p) level
of theory. Black color represents carbon atoms, blue color represents hydrogen atoms, red color
represents oxygen and green color represents Cl atoms.





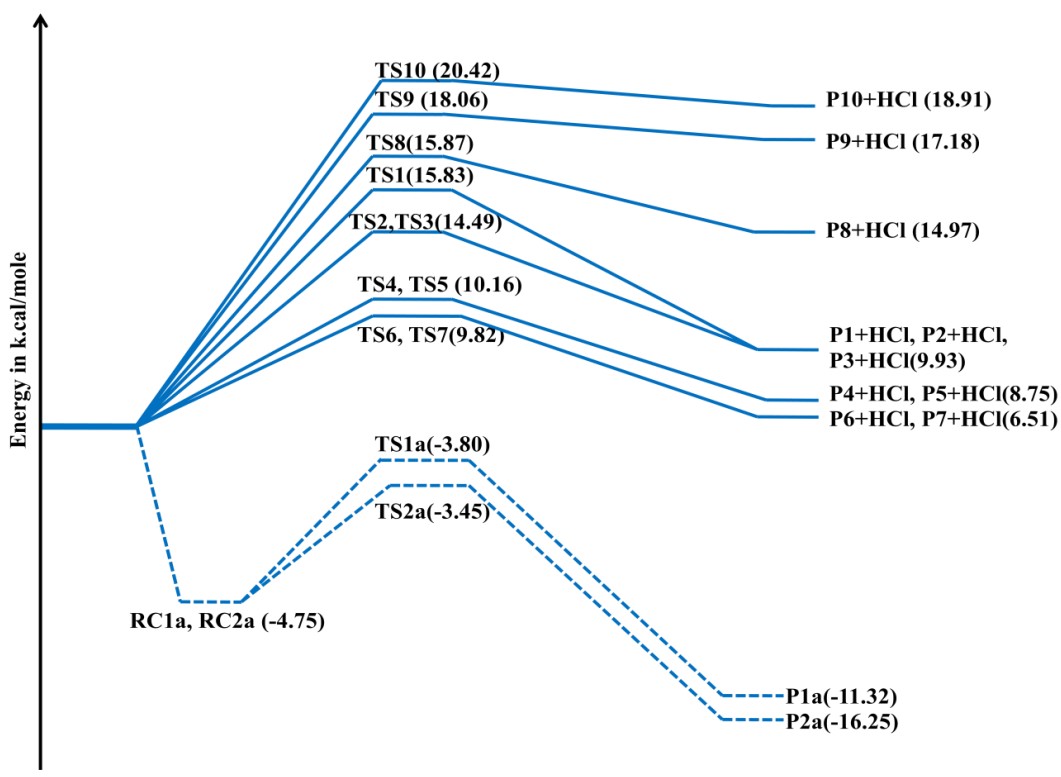

2   **Figure 8**: Potential energy diagram for the reaction of Cl atoms with 5-hexen-2-one obtained at

3   CCSD(T)/6-31+G(d,p)// MP2/6-311++G(d, p) level of theory at 298K.

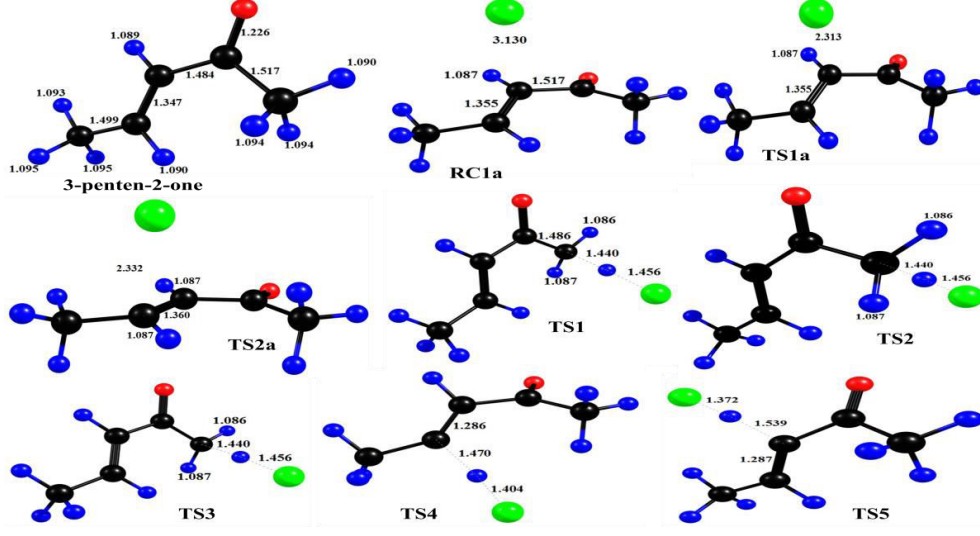





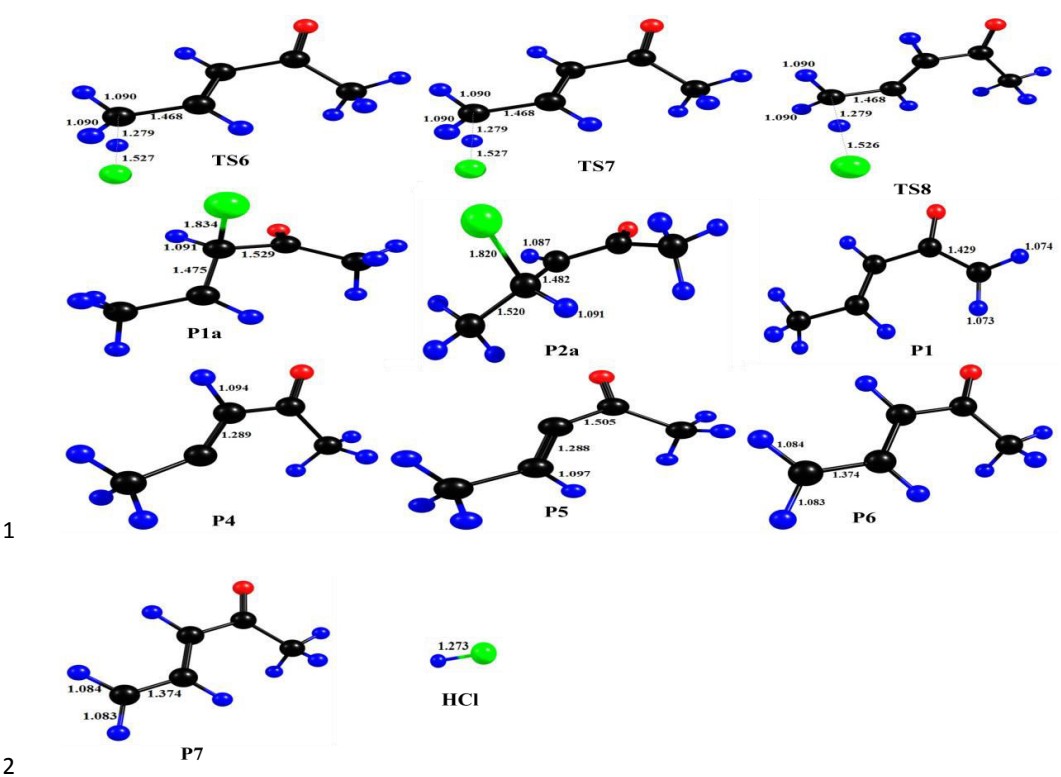

**Figure 9:** Optimized geometries of the reactants, pre-reactive complexes, transition states and
products obtained MP2/6-311++G(d,p) level of theory. Black color represents carbon atoms,
blue color represents hydrogen atoms, red color represents oxygen and green color represents Cl
atoms.




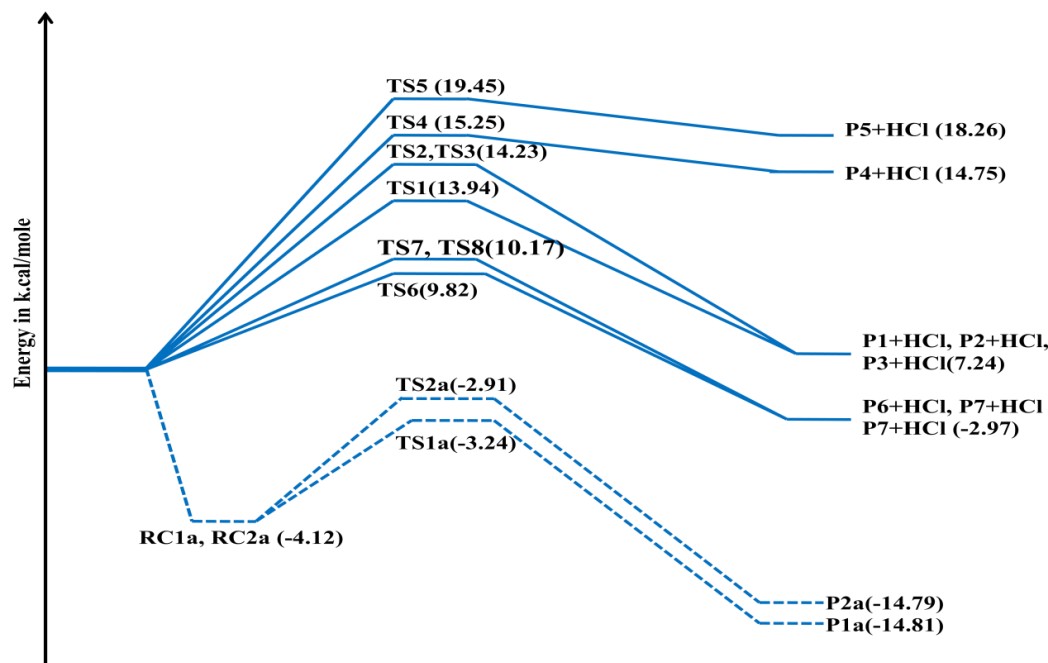

**Figure 10**: Potential energy diagram for the reaction of Cl atoms with 3-penten-2-one obtained at
CCSD(T)/6-31+G(d, p)// MP2/6-311++G(d, p) level of theory at 298K.

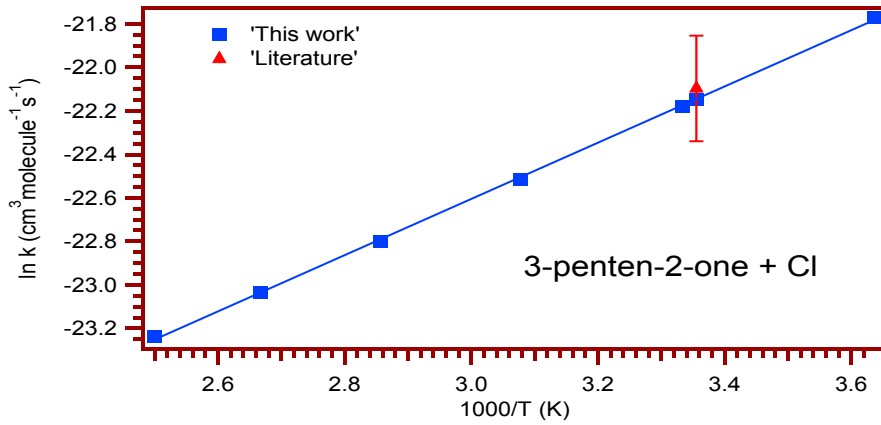

**Figure 11**: Arrhenius plot of CVT/SCT rate coefficients obtained at the CCSD(T)/6-31+G(d, p)//
MP2/6-311++G(d,p) level of theory between the temperatures 275 and 400 K for the reaction of
Cl atoms with 3-penten-2-one.



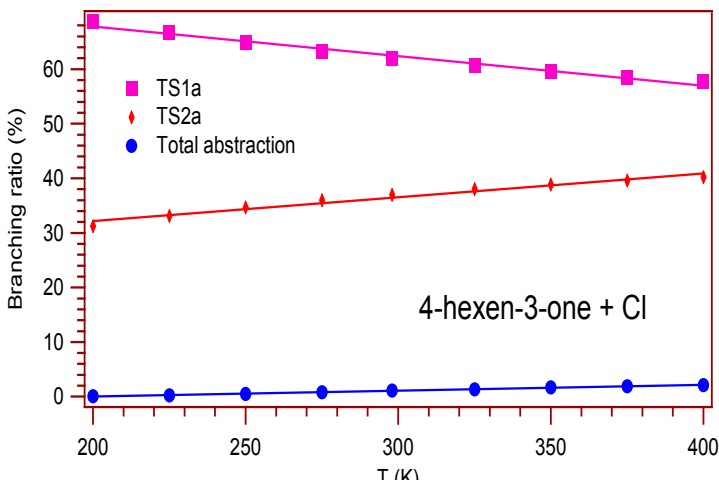

**Figure 12:** Calculated branching ratios vs temperature for the reaction of Cl atoms with 4-hexen-
3-one.

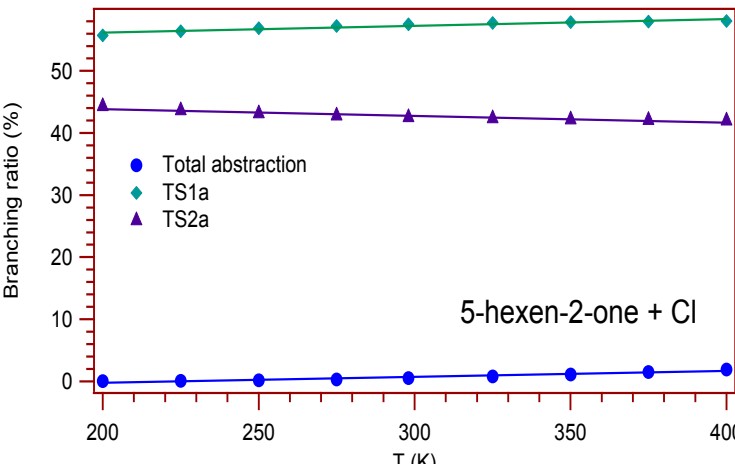

**Figure 13:** Calculated branching ratios vs temperature for the reaction of Cl atoms with 5-hexen-
2-one.





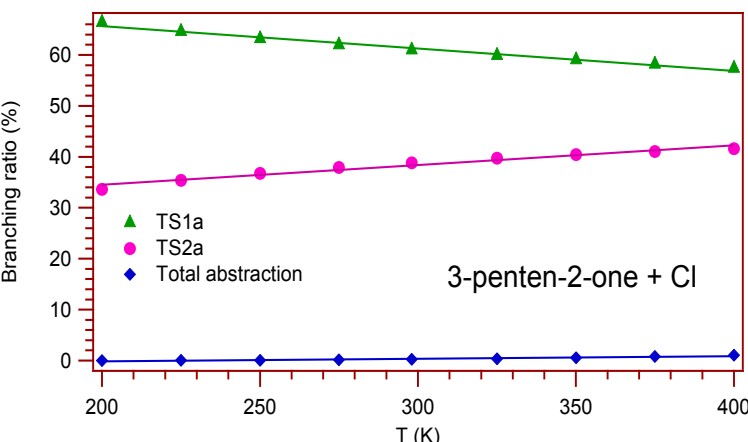

**Figure 14:** Calculated branching ratios vs temperature for the reaction of Cl atoms with 3-
penten-2-one.




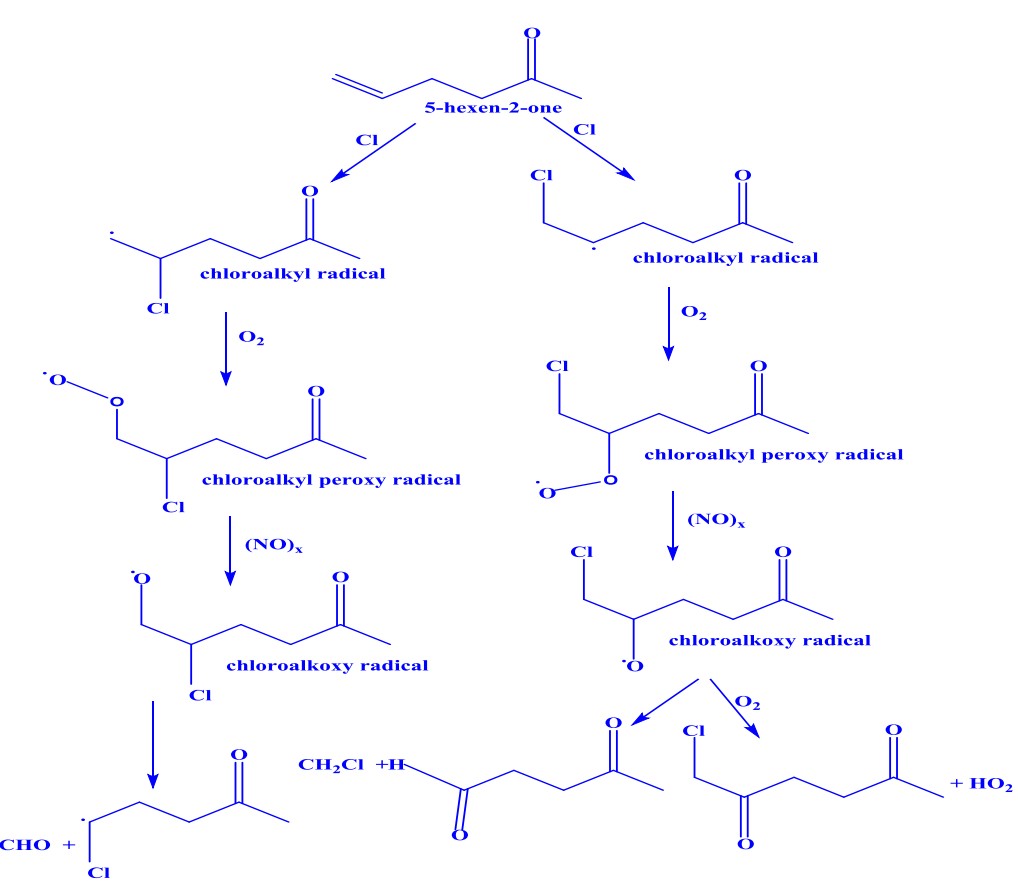

3    **Figure 15:** Atmospheric degradation mechanism for the reaction of Cl atom with 5-hexen-2-one.

