# Peer review of "Manuscript under review for journal Atmos. Chem. Phys."

_Atmospheric Chemistry and Physics, 2017_

## Referee Comment (RC1) · Anonymous Referee #1 · 8 May 2017

General Comments: Authors have reported experimental and computational studies on gas phase reaction kinetic of Cl atom + unsaturated ketones. These species have been detected in the atmosphere. The paper describes experimental and computational rate coefficients, which can be helpful to understand the reactions mechanism of unsaturated ketones initiated by Cl radical. The theoretical and experimental rate coefficients were compared with data available in the literature. It is an interesting paper. This paper is publishable after minor revisions noted.

Specific Comments: The unsaturated ketones 4-hexen-3-one, 5-hexene-2-one and 3-

penten-2-one have E and Z isomers. I don't see any discussions on their isomers. Neglecting these isomers may lead error in the computational k, which is near about factor of 2 at room temperature. Maybe Hindered rotor treatment can solve the discrepancy of the computational rate with the reference rate (experimental). I can see in Figure 2, the rate coefficient agree within 50% with the reference value and previously measured value. However, the calculated rate coefficient at the lower temperature ($\sim$275K) is near about a factor of 10 and at the higher temperature (400K) it is $\sim$100 lower than the experimental value. In my experience, this problem could be due to HR treatment and neglecting the other isomer. Another way to improve the calculations is to check MP2 imaginary frequencies. These errors are related to pre-exponential factor.

A little adjustment of barrier heights maybe gives you closer value in the entire temperature range. You can find the error in the energy calculation based previous papers JPC A J. Phys. Chem. A 2015, 119, 7578$-$7592 and J. Phys. Chem. A 2016, 120, 7060$-$7070. You can discuss your results similar way and cite these papers.

Minor Comments:

Abstract No need to write a full description of Arrhenius expressions. Maybe something like k1 = ——, k2 =. . ..will work.

Introduction: Page 1 Line 2: I think hydrocarbon should be written as an Organic Compounds. Hydrocarbon contains only hydrogen and carbon. Line 5-7: Need citations. Line 17-19: these ketones, please be specific. Page 5 line 11: Why rate expression is in the bold letter? Need equation number.

Computational Methodology Page 1 Line 1: MP2 calculations are based on full electrons or based on frozen cores? Page 1 line 3: What are the reactive complexes (RCs)? I think it should be PRC (Pre-reactive Complex). Page 1 line 5: IRC calculation at MP2 level for what reason?

Results and Discussion Page 7 Line 3: "Maybe this is due to the differences in the rate coefficients of the reference compounds and uncertainties associated with the reference compounds which were used in the present measurements. This statement confused me, whether your measured rate coefficients are correct or computational rate coefficients? Need modifications.

Figure 3: For H- atom, I think, white or gray color is appropriate. I think Blue color represent N atoms.

Figure 3: TS1 and TS2a should be corrected as discussed in Page 8 in structure 1 4-hexen-3-one.

Figure 4. PES is incomplete; I don't see the PRC energy. I guess the Electronic + ZPE is calculated at 0 K. Not sure that if authors have included thermal corrections?

Page 9: I think Products P1a and P2a should be corrected as Intermediates or Adults.

Page 9: Why not comparing your theoretical value to your experimentally measured value for all three cases in the same place where you compare with Blanco et al.

Figure 8 and 9: Structure of RC2a is missing. If RC1a and RC2a are structurally and energetically same, then why two different pre-reactive complexes?

Page 11: Again this statement should be corrected "Maybe this is due to the uncertainties associated with the submerged transition states. I don't think submerged TS can underestimate or overestimate the k. Maybe problem-related to and theoretically calculated pre-exponential factors or calculated energies using at MP2 level. Please refer to my earlier comments.

I don't think the value reported -3.80 and -3.45 kcal/mol are the barrier heights for addition reactions. Check it.

Table 9: Something wrong with title or table contents.

Conclusions:

Authors stated in the conclusion "As these molecules are short-lived they would not contribute to global warming in any time horizons." Then why there are performing the measurement and calculations? Some benefit should be added. Also, why reporting lifetimes at 298K.

---

## Referee Comment (RC2) · T. S. Dibble (Referee) · 9 May 2017

Review of Cl + alkenenones for ACPD by Theodore S. Dibble (tsdibble@esf.edu)

GENERAL COMMENTS

This manuscript reports relative rate experiments and computations on the kinetics of Cl reacting with a series of unsaturated ketones as a function of temperature. These compounds are of some interest and the reaction of Cl with these compounds can contribute significantly to their loss under some atmospheric conditions. The experiments (on two compounds) appear to be solid, although additional details should be reported and the clarity improved. Also, there is no experimental rate constant data on the reference compounds at the higher temperatures studied here. As a result, the absolute rate constants (as opposed the relative rate constants) at those higher temperatures may not be reliable.

The clarity and precision of the writing should be improved.

The computational chemistry calculations are not reliable. While it the authors could address issues with the energeties obtained from quantum chemistry, it is probably not feasible to reliably compute rate constants and branching ratios for the systems being studied here.

I am uncertain as to the importance of the two compounds being studied experimentally, and the experiments only provide rate constants, not branching ratios. So the manuscript remaining after removing the computational parts probably does not belong in ACP.

SPECIFIC COMMENTS ON EXPERIMENTS

1.     Secondary chemistry appears to be a minor factor, but the reference compounds are sufficiently similar that they might give rise to similar secondary chemistry. It would help to present a brief discussion about WHY secondary chemistry is expected be minor. Just in case someone wants to model the experiment, the manuscript should include the repetition rate of the 248 nm lasers and its fluence (in mJ cm$^{-2}$ pulse$^{-1}$ rather than mJ pulse$^{-1}$). For the same reason, the initial concentration range of the test compounds and reference compounds should be listed in the text.

2.     An expanded version of Table 1 is needed that includes the initial concentration of the reference compound. Also, results should be presented with experiments with $O_2$ present.

Ideally, the Supplementary Material would include the measured concentration of the test compounds and the reference compound at 1200, 1400, 1600, 1800 and 2000 pulses for each experiment.

3.     The literature values of the rate constant for Cl + isoprene and Cl + 1-pentene are only known up to 320 K. There is no reason to expect Arrhenius behavior from these reactions. Consequently, it is not appropriate to derive absolute rate constants at these temperatures without highlighting the fact that rate constants for the reference compounds are being extrapolated.

4.     The value of [Cl] used to compute atmospheric lifetimes (in Table 10) is only valid in a small part of the atmosphere. This should be noted.

SPECIFIC COMMENTS ON CALCULATIONS

1.      The CCSD(T) energy calculations use a basis set that is far too small to be reliable. This is evident in many of the reported values of the critical energies for hydrogen abstraction (they are far too high). Could there also be a problem with unstable wavefunctions contributing to high energies of these TSs? The small basis set could also (via basis set superposition error) lead to TS energies for the addition reaction that are lower than the actual values. The fact that the reported rate constants (dominated by addition) agree with experiment is due to fortuitous cancellation of error. A basis set extrapolation scheme or composite method is needed for accurate treatment of both types of TSs.

2.      The addition of Cl to the test compounds to form a reactive complex must have a variational TS (in addition to the saddle point reported for the formation of a covalent chlorine-carbon bond). Strictly speaking, a 2-TS approach is needed to obtain the rate constant in the high-pressure limit (see J. Phys. Chem. A, 2006, 110, pp 6960–6970), but I would not insist on authors carrying out these calculations.

        The manuscript implicitly assumes the addition reactions are in the high pressure limit at 1 atm over the temperature range reported. This may be reasonable, but should be stated explicitly.

3.      There is an issue with the computations that the authors won't be able to overcome: the general approach used here probably has limited applicability to Cl reaction with molecules containing C=C double bonds. See the 2014 paper by A.G. Suits and A. M. Mebel (DOI: 10.1038/ncomms5064). This paper makes two major points relevant here:

        a) H-abstraction from allylic sites proceeds without a barrier (3-penten-2-one and 4-hexen-3-one have allylic sites).

        b) There exists roaming paths connecting chlorine adducts of the alkenes to HCl formation (potentially relevant to all three species studied in the manuscript under review).

        The results in this paper have been verified and extended. The reaction paths described in (a) and (b) are important under the conditions of the experiment, and (a) is important. Although those conditions are far different than those in the atmosphere, another paper (DOI: 10.1038/srep40105) suggests that roaming paths leading to HCl are more important at thermal energies (meaning atmospheric conditions) than the conditions of the Suits and Mebel paper. This conclusion

        The results found in this manuscript are not consistent with those of the Suits and Mebel paper or subsequent computations. The roaming paths can only be treated by running dynamics, and that is far outside the scope of this manuscript. As a result, it appears that reliable kinetic insight cannot be obtained from the general approach used in the present manuscript.

        For those who got lost in the gory details of the physical chemistry I just discussed, just know that the theoretical approach in the present manuscript is, at least in part, inconsistent with obtaining the ~17% branching percentage (at 298 K) found for H-abstraction in the reaction of Cl + isoprene (Bedjanian et al., 1998, cited in the manuscript).

4.      As Anonymous Referee #1 points out, hindered rotor corrections to rate constants may be important.

TECHNICAL CORRECTIONS

1.    The GC temperature program and flow rate should be specified (at least in the Supplementary Material).

2.    On page 5 where the absence of loss of test and reference compounds was verified in the dark and in the absence of oxalyl chloride, please specify the upper limit to the loss (e.g., < 4%). Similarly, specify the upper limit to the change in rate constant upon adding $O_2$ (and the partial pressure of $O_2$ used).

3a.    The sample of 4-hexen-3-one is listed as >90% trans. The manuscript should specify whether the reported GC measurements were only of the trans isomer. Also, the manuscript should specify the cis/trans composition of 4-hexen-3-one as it appears in nature (if known). Every time the description of the calculations identifies the test compounds, they should specify trans (e.g., "*trans*-4-hexen-3-one" rather than "4-hexen-3-one").

3b.    Is the 5-hexen-2-one used in experiments all trans?

4.    In the computational methodology, expand the acronyms CVT and SCT. The partition function of the reactant does not depend on *s*, as stated here. $V_{MEP}$ should be specified as a potential energy **difference** (corrected for zero-point energy). The value of the reaction path degeneracy for each TS should be specified somewhere.

5.    In Table 10, specify whether the experimental or theoretical rate constants used for Cl reactions with 4-hexen-3-one and 5-hexene-2-one. Also, add a second digit to the lifetimes with respect to reaction with OH, and only use two significant figures elsewhere.

6.    Caption to Table 11. These are "percentages" not "ratios".

7.    Conformers:

    - Did the authors choose the conformers because they were the minimum energy conformers? If so, what efforts were made to verify this?

    - Two of the three test compounds are listed as having near-Cs symmetry, and probably should be treated as having Cs symmetry. This means that there are fewer unique transition states than listed (e.g., for 4-hexen-3-one TSs 8-10 are only two unique TSs).

8.    Can Cl form a van der Waals complex with the $\pi$ cloud of the carbonyl groups?

9.    Both "test" and "sample" are used for the alkenones; please standardize terminology

10.    In Table 1, some of the error bars don't make sense, e.g., bottom of page 21 (298 K):
    - for 1,3 butadiene as a reference, the error bar on $k_{test}/k_{ref}$ is 25%, but the error bar on $k_{test}$ is only 7%.
    - for isoprene as a reference, the error on the three individual values of $k_{test}/k_{ref}$ is on the order of 7-9%, and that should be reflected in the average value of $k_{test}/k_{ref}$.

    - for the final value of $k_{test}$ averaged over multiple reference compounds, the error bar should be closer to 1.0 than 0.4 ($\times 10^{-10}$ $cm^3$ molecule$^{-1}$ sec$^{-1}$).

11.    The Supporting Information is more complete than many, but it should also include absolute energies at 0 K, zero-point energies, and (ideally) H and G at 298 K. Also add

the CVT rate constant and tunneling corrections versus temperature for each reaction path for all three test molecules.

12. The Introduction does not reflect a thorough understanding of atmospheric chemistry and cites too few recent papers.

13. Page 6: use a lower case rather than upper case kappa for tunneling corrections.

14. The equations given for rate constants on page 6 have units of $sec^{-1}$. Please correct them.

15. On page 7, the results of Bedjanian et al. were at low pressure, not atmospheric pressure, although the rate constant was reported to be independent of pressure.

16. On page 7, lines 28 "10%" should be "16%"

---

## Referee Comment (RC3) · V. C. PAPADIMITRIOU (Referee) · 18 May 2017

**Manuscript:** acp-2017-163

**Authors:** S. Vijayakumar, Avinash Kumar and B. Rajakumar

**Ms. Title:** Experimental and computational kinetics investigations for the reactions of Cl atoms with series of unsaturated ketones in gas phase

In the present work, the authors report experimentally and theoretically obtained rate coefficients for the reaction of Cl atoms with three unsaturated ketones, i. e., $CH_3CH=CHC(O)C_2H_5$, $CH_2=CH(CH2)_2C(O)CH_3$ and $CH_3CH=CHC(O)CH_3$ that are abundant in the atmosphere, employing relative rate methods and ab-initio calculations (CCSD(T)/6-31+G(d,p)//MP2/6-311++G(d,p)), respectively. In addition to that, the temperature dependence of the title reactions was also studied, both experimentally (298 – 363 K) and theoretically (275 – 400 K), at atmospheric pressure. Finally, ab-initio calculation were also used to facilitate reaction mechanism investigation.

The present reviewer believes that the present study does fit in ACP, but there are several, both major and minor issues that the authors need to address before the current submission would be in a publishable form.

**Minor issues:**

1. All the sentences that start with *witch* and *where* should include a comma before that, i.e., ,*which*, throughout the manuscript.

2. Keywords should be one word

3. **Pg 2. line 9, Introduction:** Please change *effects* with *affects*.

4. **Pg 2. line 12, Introduction:** Please rephrase *abundant attention*

5. **Pg 3. line 1, Introduction:** Please change *different* with *the dominant*

6. **Pg 3. line 6, Introduction:** Please change *reported them to be* to *the reported values were*

7. **Pg 3. line 16 Introduction:** Please note that data is a count noun when you refer to experimental data and through the whole text you should change the syntax of *data is* with *data are*.

8. **Pg 5. line 22 Chemicals:** Please replace *freeze-pump-thaw* with *freeze-pump-thaw cycle*

9. In the reaction scheme the temperature range given is wrong (200 – 400 K). Please correct accordingly.

There are several similar issues that the authors should address in the whole paper. It is strongly recommended fresh eyes should read through the whole text.

**Major issues:**

**Title:**

Among the three test compounds studied in the present work, only two were looked into experimentally, which probably mean that the word series should be omitted from the paper title.

1. Include the pressure conditions and the bath gas (Include those conditions in tables as well) used in the majority of the experiments in Abstract section.

2. Authors report uncertainties in the abstract and in all other sections of the paper, tables etc., but there is no explanation of how were they determined and what they represent. Do they include systematic uncertainties and if they do, how were they estimated? An error analysis section will assist the reader to go through the paper and assess the reliability of the measurements.

3. In general, the present reviewer believes that the submitted draft needs severe reorganisation as theoretical calculations and experimental results for each ketone have substantial similarities. It would be more meaningful to present experimental results for both ketones and they should be separate from the corresponding theoretical ones, to avoid repetitions. Then, the authors might include a section where all the important observations would be summarized and comparisons between experiments and theory, relative positioning of the double bond and carbonyl group (experiment vs experiment and theory vs theory) should be highlighted and interpreted in terms of reactivity.

4. Although $CH_3CH=CHC(O)CH_3$ vapour pressure is low, it should be adequate for relative rate measurements, especially at higher temperatures. Experiments, even at those conditions would assist to understand how the relative positioning of the double bond and carbonyl group affects unsaturated ketones reactivity, as well as what the impact to the compete reaction mechanisms of abstraction and association is. It might also be of worth to try to do some experiments with the title ketone, especially at high temperatures.

5. Regarding theoretical calculations: a. How the authors verified that they have located global minima on the PES, during geometries optimisation. b. Although the authors employed a level of theory of high accuracy (CCSD(T)), the basis set used was unexpectedly limited. Why did the authors chose a non-correlation consistent, double-zeta basis set for single point calculations? c. Although it is reasonable, the authors should clearly state that they have also calculated ZPE using MP2 method and that they have calculated their values at 296 K.

6. Reference reactions were measured in a narrower temperature range compared to the studied reactions. This is to say that the authors have assumed that reaction mechanism does not change as a function of temperature. This might be partly true and safe when reactions proceed only via –H abstraction, which is not the case here, where abstraction and association compete to each other and their contribution is expected to vary with temperature. It is strongly recommended the authors to avoid such assumptions and hypothetical extrapolations. It would have also been more appropriate to use as references, reactions that the rate coefficient is not pressure dependent. On the top of that, the authors recognize that pressure dependent measurements would assist to understand the mechanism of the reaction and they will reveal how the competition between the pressure independent channels of H abstraction competes with the pressure dependent pathway of Cl association to the double bond. From an atmospheric perspective, those very short lived compounds (VSLC) will not be substantially transferred to the upper troposphere ad thus rate coefficient measurements at atmospheric pressure are the most relevant ones with regard to the atmospheric impact.

7. Mechanism was not thoroughly investigated and no critical SOA formation intermediates were identified. The part that the authors try to interpret the mechanism in terms of electronic effects due to the presence of carbonyl group is incomplete and steric factors and entropy change to the TS should be also considered. A comparative plot of the abstraction and association kinetics as a function of temperature would assist to elucidate the relative importance.

**Tables:**

1. Relative rate tables: In all cases ratios uncertainties and bimolecular rate coefficients ones are identical, which is to say that the authors have not propagated the error limits. Revision is required through the whole draft. (e.g., Table 1. r = 1.46 **±0.36** and k =(5.08

±**0.36**) ×10$^{-10}$ cm$^3$ molecule$^{-1}$ s$^{-1}$). The sources of the quoted uncertainties should also be included.

2. **Table 6.** The authors report a rate coefficient of 1.09 ×10$^{-9}$ cm$^3$ molecule$^{-1}$ s$^{-1}$ at 275 K that is higher than the collision rate limit (~8 ×10$^{-10}$ cm$^3$ molecule$^{-1}$ s$^{-1}$ at room temperature). This is possible only when reaction follows a Harpoon mechanism, which means ion-ion or ion-molecule interaction, which definitely is not the case here. Most likely, this is due to intrinsic problems of the computational methods and if there was an estimated uncertainty of the calculated k values via benchmarking calculations, the error limits would have revealed that this is due to the limitations of the theoretical methods employed.

**Graphs:**

1. All the relative rate plots should begin from zero difference in test and reference concentration relative variance (0,0). Negative values are meaningless. Moreover, figure captions should include all the experimental conditions and explain everything that is shown inside the plots and in insets of the graph. Finally, for the shake of clarity, please use different symbols when different references are included in the graph (Not only different colors).

2. In Arrhenius plots, first, include an X-mirror axis so as the reader to have a measure of temperature. Also, use k (and log axis) in Y-axis. Note that lnk has no units and if the authors decide to keep it this way, they should refer to it as ln(k, cm$^3$ molecule$^{-1}$ s$^{-1}$)

3. In some cases, curvature was observed in Arrhenius plot. Why didn't the authors fit their data using a modified version of Arrhenius fitting (A ×T$^n$), which include temperature dependence on the pre-exponential factor and is quite common in association reactions.

**Conclusions:**

1. Conclusions are very limited and not appropriate for such an amount of work.

2. GWP contribution is negligible, since the studied ketones contain no C-F bonds and more importantly they are very short lived compounds. Did the authors look if there are any available IR spectra in the literature that they can comment on them to justify their statements?

3. The authors could estimate a POCP for the studied ketones and compare with similar unsaturated compounds. This is expected to be the major contribution of those compounds on air-quality issues.

4. Cl chemistry importance should be commented in conclusion part. Although it is not expected to substantially affect the fate of ketones, since they are extremely short lived compounds, it might be of importance if chlorinated products are formed. In particular for such short lived compounds it might also affect their POCP, taking into account that metropolitan cities and polluted areas are responsible for the huge majority of emissions of such compounds and in most of the cases they are located near by the sea, where the higher levels af Cl atoms are observed.

---

## Author Comment (AC1) · 19 Jul 2017

We thank the reviewer for going through the manuscript entitled "Experimental and computational kinetics investigations for the reactions of Cl atoms with unsaturated ketones in gas phase" and for his/her constructive suggestions to improve the quality of the manuscript. We have incorporated all the suggestions and given explanations to the queries in the revised manuscript (RMS) at appropriate places.

[Figure]

The changes/additions in the revised manuscript are given in blue color for ready reference. The complete rebuttal is given below. Anonymous Referee #1 General Comments: Authors have reported experimental and computational studies on gas phase reaction kinetic of Cl atom + unsaturated ketones. These species have been detected in the atmosphere. The paper describes experimental and computational rate coefficients, which can be helpful to understand the reactions mechanism of unsaturated ketones initiated by Cl radical. The theoretical and experimental rate coefficients were compared with data available in the literature. It is an interesting paper. This paper is publishable after minor revisions noted. Specific Comments: The unsaturated ketones 4-hexen-3-one, 5-hexene-2-one and 3-penten-2-one have E and Z isomers. I don't see any discussions on their isomers. Neglecting these isomers may lead error in the computational k, which is near about factor of 2 at room temperature. May be Hindered rotor treatment can solve the discrepancy of the computational rate with the reference rate (experimental). I can see in Figure 2, the rate coefficients agree within 50% with the reference value and previously measured value. However, the calculated rate coefficient at the lower temperature (275K) is near about a factor of 10 and at the higher temperature (400K), it is 100 lower than the experimental value. In my experience, this problem could be due to HR treatment and neglecting the other isomer. Another way to improve the calculations is to check MP2 imaginary frequencies. These errors are related to pre-exponential factor. A little adjustment of barrier heights may be gives you closer value in the entire temperature range. You can find the error in the energy calculation based previous papers J. Phys. Chem. A 2015, 119, 7578−7592 and J. Phys. Chem. A 2016, 120,7060−7070. You can discuss your results similar way and cite these papers. Response: In both 4-hexen-3-one and 3-penten-2-one one methyl group is attached at the terminal carbon of the double bond. Therefore, E (trans) and Z (cis) isomers exist for both the compounds. Whereas, in case of 5-hexen-2-one, the double bond is connected to three hydrogens (two at the terminal carbon and one at the other end of the double bond) and therefore it exists as a

single conformer only. We tried to optimize both trans and cis isomers of the first two systems. However, we could not optimize the cis isomers, probably as they are highly unstable. Under these circumstances, we have considered the most stable trans isomer in our calculations. Hindered rotor (HR) calculations were performed and compared with the present experimental and reported rate coefficients (given below). Rate coefficients obtained including HR corrections are almost equal to our earlier theoretical calculations. As the reviewer rightly pointed, the discrepancy between theoretical and the present experimental rate coefficients may be due to the errors in pre-exponential factors and the errors in the estimation of barrier heights. As the rate coefficient at a given temperature is the combination of both pre-exponential factor and the activation energy, the difference can be attributed to the accuracy with which both these factors are determined. The pre-exponential factor depends on how best the partition functions of reactants and transition states are estimated, which in turn depends on the vibrational frequencies obtained in the calculations. On another hand, the uncertainties in the calculated energies of transition states can critically affect the calculated rate coefficients. Lynch et al., 2001;Ali et al., 2016; and Ali et al., 2015 concluded that, there would be an error of about 1.1 kcal mol-1 in the barrier height calculations at the CCSD(T) level of theory with 6-31+G(d,p) basis set. The same level of theory and the basis set were used in the present calculations. Therefore, given an uncertainty of about 1 kcal mol-1 in the activation barrier, the theoretically calculated rate coefficients are in reasonable agreement with the reported experimentally measured ones. This discussion is added in the RMS. Table: Comparison of the rate coefficients (cm3 molecule-1 s-1) for the reactions of unsaturated ketones with Cl atoms at 298K. 4-hexen-3-one + Cl 5-hexen-2-one + Cl 3-penten-2-one + Cl k Theory 3.66×10-10 5.56×10-10 2.4×10-10 k Theory with HRcorrection 3.60×10-10 5.47×10-10 2.38×10-10 k Experimental (5.55±1.31)×10-10 (4.14±1.25)×10-10 - k Blanco et al. (3.00±0.58)×10-10 (3.15±0.50)×10-10 (2.53±0.54)×10-10 Table: Comparison of the theoretically obtained rate coefficients (cm3 molecule-1 s-1) for the reaction of Cl atoms with unsaturated ketones at CCSD(T)/6-31+G(d, p)//MP2/6-311++G (d, p) level

of theory over the temperature range of 275-400K. 4-hexen-3-one + Cl 5-hexen-2-one + Cl 3-penten-2-one + Cl T (K) k Theory k Theory with HRcorrection k Theory k Theory with HRcorrection k Theory k Theory with HRcorrection 275 $5.81\times10^{-10}$ $5.73\times10^{-10}$ $1.09\times10^{-09}$ $1.07\times10^{-09}$ $3.51\times10^{-10}$ $3.48\times10^{-10}$ 298 $3.66\times10^{-10}$ $3.60\times10^{-10}$ $5.56\times10^{-10}$ $5.47\times10^{-10}$ $2.40\times10^{-10}$ $2.38\times10^{-10}$ 325 $2.33\times10^{-10}$ $2.29\times10^{-10}$ $3.85\times10^{-10}$ $3.78\times10^{-10}$ $1.66\times10^{-10}$ $1.65\times10^{-10}$ 350 $1.64\times10^{-10}$ $1.62\times10^{-10}$ $2.44\times10^{-10}$ $2.40\times10^{-10}$ $1.26\times10^{-10}$ $1.24\times10^{-10}$ 375 $1.22\times10^{-10}$ $1.20\times10^{-10}$ $1.57\times10^{-10}$ $1.54\times10^{-10}$ $9.89\times10^{-11}$ $9.79\times10^{-11}$ 400 $9.49\times10^{-11}$ $9.35\times10^{-11}$ $1.00\times10^{-10}$ $9.83\times10^{-11}$ $8.07\times10^{-11}$ $7.99\times10^{-11}$ Lynch, B. J. and Truhlar, D. G.: How well can hybrid density functional methods predict transition state geometries and barrier heights? J. Phys. Chem. A 105, 2936-2941, 2001. Ali, M. A., Sonk, J. A. and Barker, J. R.: Predicted chemical activation rate constants for HO2 + CH2NH: The dominant role of a hydrogen bonded pre-reactive complex. J. Phys. Chem. A 120, 7060-7070, 2016. Ali, M.A. and Barker, J.R.: Comparison of there isoelectronic multiple-well reaction systems: OH+CH2O, OH+CH2CH2, and OH+CH2NH. J. Phys. Chem. A 119, 7578-7592, 2015. Minor Comments: Abstract No need to write a full description of Arrhenius expressions. May be something like k1 = ——, k2 =. . ..will work. Response: It was corrected and Arrhenius expressions are written as k1= $(2.82\pm1.76)\times10^{-12}$exp [(1556$\pm$438)/T] cm3 molecule-1 s-1 and k2 = $(4.6\pm2.4)\times10^{-11}$exp[(646$\pm$171)/T] cm3 molecule-1 s-1 in the RMS. Introduction: Page 1 Line 2: I think hydrocarbon should be written as an Organic Compounds. Hydrocarbon contains only hydrogen and carbon. Line 5-7: Need citations. Response: hydrocarbons word is replaced with organic compounds. The following reference is cited for lines 5-7 in the RMS. Seinfeld, J. H. and Pandis, S. N.: Atmospheric Chemistry and Physics: From Air Pollution to Climate Change; John Wiley & Sons: New York, 1997. Line 17-19: these ketones, please be specific. Response: ketones are specified in RMS and they are acetone, 2-butanone, 3-pentanone and 2- pentanone. Page 5 line 11: Why rate expression is in the bold letter? Need equation number. Response: Bold fonts are now replaced with regular fonts and all equations are numbered in the RMS. Computational

Methodology Page 1 Line 1: MP2 calculations are based on full electrons or based on frozen cores? Response: These MP2 calculations are based on frozen cores. Page 1 line 3: What are the reactive complexes (RCs)? I think it should be PRC (Pre-reactive Complex). Response: They are Pre-reactive Complexes (PRCs) and corrected in RMS. Page 1 line 5: IRC calculation at MP2 level for what reason? Response: Intrinsic Reaction Coordinates (IRCs) calculations were performed to check the transition states are connected to reactant and products. Also, to find if all the reaction pathways are independent of each other. This is added in the computational methodology of the RMS. Results and Discussion Page 7 Line 3: "May be this is due to the differences in the rate coefficients of the reference compounds and uncertainties associated with the reference compounds which were used in the present measurements. This statement confused me, whether your measured rate coefficients are correct or computational rate coefficients? Need modifications. Response: The above sentence is modified as "The difference between the present experimental and reported rate coefficient, may be due to the differences in the rate coefficients of the reference compounds and associated uncertainties, which were used in the present measurements" in the RMS. Figure 3: For H- atom, I think, white or gray color is appropriate. I think Blue colorrepresent N atoms. Response: The standard colors for 'C' atom - gray color, 'H' atom –white color, 'O' atom- red color and 'Cl' atom – green color are used in the RMS. Figure 3: TS1 and TS2a should be corrected as discussed in Page 8 in structure 1: 4-hexen-3-one. Response: Addition transition states are represented by TS1a (adjacent to the CH3 group) and TS2a (adjacent to the C=O group) as shown in the structure 1: 4-hexen-3-one in the RMS. Figure 6. PES is incomplete; I don't see the PRC energy. I guess the Electronic + ZPE is calculated. Not sure that if authors have included thermal corrections? Response: We have incorporated PRCs with energies in Figure 6 (PES) in the RMS. We have taken the sum of electronic and zero-point energies. We have not included thermal corrections. Page 9: I think Products P1a and P2a should be corrected as Intermediates or Adults. Response: As the products P1a and P2a are alkyl radicals, we have renamed as intermediates

in the RMS as given below. "The Cl atom addition on double bond at TS1a and TS2a lead to the formation of intermediates Pla and P2a respectively". Page 9: Why not comparing your theoretical value to your experimentally measured value for all three cases in the same place where you compare with Blanco et al. Response: We have reorganized experimental and theoretical sections in the RMS, which gives the better continuity and explain all three systems in the same place. Figure 7 and 9: Structure of RC2a is missing. If RC1a and RC2a are structurally and energetically same, then why two different pre-reactive complexes? Response: In Figure 7 and 9, structures of RC1a and RC2a are energetically same but their structures are different. The Cl atom addition at TS1a and TS2a leads to the formation of prereactive complexes via PRC1a and PRC2a respectively and these structures are given in the RMS. Page 11: Again this statement should be corrected "May be this is due to the uncertainties associated with the submerged transition states. I don't think submerged TS can underestimate or overestimate the k. May be problem-related to and theoretically calculated pre-exponential factors or calculated energies using at MP2 level. Please refer to my earlier comments. Response: We have modified the above sentence as "the discrepancy between theoretical and the present experimental rate coefficient may be due to the errors in pre-exponential factors and the errors in the estimation of barrier heights. As the rate coefficient at a given temperature is the combination of both pre-exponential factor and the activation energy, the difference can be attributed to the accuracy with which both these factors are determined. The pre-exponential factor depends on how best the partition functions of reactants and transition states are estimated which in turn depends on the vibrational frequencies obtained in the calculations." This discussion is added in the RMS. I don't think the value reported -3.80 and -3.45 kcal/mol are the barrier heights for addition reactions. Check it. Response: For the reaction of Cl atom with 4-hexen-3-one, the relative energy barrier heights for TS1a and TS2a are -3.80 and -3.45 kcal mol-1 are correct and given in Table 6 of the RMS. Table 9: Something wrong with title or table contents. Response: In Table 9, we have compared the rate coefficients for the reactions of unsaturated

ketones with OH radicals, Cl atoms and NO3 radicals. The title of the table is modified as "Comparison of the rate coefficients for the reactions of unsaturated ketones with Cl atoms, OH and NO3 radicals at 298K" in the RMS. Conclusions: Authors stated in the conclusion "As these molecules are short-lived they would notcontribute to global warming in any time horizons." Then why there are performing themeasurement and calculations? Some benefit should be added. Also, why reportinglifetimes at 298K. Response: The cumulative lifetimes of the studied molecules are estimated to be an hour and global warming potential of these molecules are negligible (0.01). When these molecules are released into the atmosphere, within an hour they are lost from its original form. And also, we have calculated ozone formation potentials in the troposphere for the title reactions, which are 7, 6 and 5 ppm for 4-hexen-3-one, 5-hexen-2-one and 3-penten-2-one respectively. The degradation of unsaturated ketones would lead to significant amount of ozone formation in the troposphere. This is added in the RMS.

Please also note the supplement to this comment:
https://www.atmos-chem-phys-discuss.net/acp-2017-163/acp-2017-163-AC1-supplement.pdf

---

## Author Comment (AC3) · 19 Jul 2017

Manuscript: acp-2017-163 Authors: S. Vijayakumar, Avinash Kumar and B. Rajakumar We thank the reviewer for going through the manuscript entitled "Experimental and computational kinetics investigations for the reactions of Cl atoms with unsaturated ketones in gas phase" and for his/her constructive suggestions to improve the quality of the manuscript. We have incorporated all the suggestions and given explanations to the queries in the revised manuscript (RMS) at appropriate places. The changes/additions in the revised manuscript are given in blue color for ready refer-

ence. The complete rebuttal is given below. Ms. Title: Experimental and computational kinetics investigations for the reactions of Cl atoms with unsaturated ketones in gas phase In the present work, the authors report experimentally and theoretically obtained rate coefficients for the reaction of Cl atoms with three unsaturated ketones, i. e., $CH_3CH=CHC(O)C_2H_5$, $CH_2=CH(CH_2)_2C(O)CH_3$ and $CH_3CH=CHC(O)CH_3$ that are abundant in the atmosphere, employing relative rate methods and ab-initio calculations (CCSD(T)/6-31+G(d,p)//MP2/6-311++G(d,p)), respectively. In addition to that, the temperature dependence of the title reactions was also studied, both experimentally (298 – 363 K) and theoretically (275 – 400 K), at atmospheric pressure. Finally, ab initio calculations were also used to facilitate reaction mechanism investigation. The present reviewer believes that the present study does fit in ACP, but there are several, both major and minor issues that the authors need to address before the current submission would be in a publishable form. Minor issues: R3Q1. All the sentences that start with witch and where should include a comma before that, i.e., ,which, throughout the manuscript. Response: We have included "comma" before which and where throughout the manuscript. R3Q2. Keywords should be one word Response: We have written as "Keyword" in the RMS. All the keywords were written in one word format.

R3Q3. Pg 2. line 9, Introduction: Please change effects with affects. Response: Corrected.

R3Q4. Pg 2. line 12, Introduction: Please rephrase abundant attention Response: We have replaced "abundant attention" with "lot of attention".

R3Q5. Pg 3. line 1, Introduction: Please change different with the dominant Response: We have replaced "different" with "dominant".

R3Q6. Pg 3. line 6, Introduction: Please change reported them to be to the reported values were Response: Corrected.

R3Q7. Pg 3. line 16 Introduction: Please note that data is a count noun when you refer to experimental data and through the whole text you should change the syntax of data

is with data are. Response: corrected throughout the RMS.

R3Q8. Pg 5. line 22 Chemicals: Please replace freeze-pump-thaw with freeze-pump-thaw cycle Response: "freeze-pump-thaw" is replaced with "freeze-pump-thaw cycle" in the RMS.

R3Q9. In the reaction scheme the temperature range given is wrong (200 – 400 K). Please correct accordingly. Response: corrected to appropriate temperatures: 298 – 363K, Experiment & 275-400K, Theory.

R3Q10. There are several similar issues that the authors should address in the whole paper. It is strongly recommended fresh eyes should read through the whole text. Response: We have gone through the whole manuscript and all grammatical mistakes and misspellings were corrected.

Major issues: Title: R3Q11. Among the three test compounds studied in the present work, only two were looked into experimentally, which probably mean that the word series should be omitted from the paper title. Response: It was removed in the RMS. After removing it, the new title is "Experimental and computational kinetics investigations for the reactions of Cl atoms with unsaturated ketones in gas phase".

R3Q12. Include the pressure conditions and the bath gas (Include those conditions in tables as well) used in the majority of the experiments in Abstract section. Response: The pressure conditions and bath gas used are given both in the abstract and the Tables 1 and 2. These two tables are appended below for the reviewer's ready reference.

Table 1: Relative rate measurements for the reaction of Cl atoms with 4-hexen-3-one over the temperature range of 298-363K relative to 1,3-butadiene, isoprene and 1-pentene. T(K) Reference compound Bath gas (Torr of O2) Pressure in Torr (ksample/ kreference) $\pm 2\sigma$ (ksample/ kreference) Average$\pm 2\sigma$ (k$\pm 2\sigma$)$\times$10-10 cm3molecule-1s-1 (k$\pm 2\sigma$)$\times$10-10 cm3molecule-1s-1 Literature (k$\pm 2\sigma$)$\times$10-10 (cm3molecule-1s-1) at 298K 298$\pm$2 1,3-butadiene N2 760 1.55$\pm$0.21 1.54$\pm$0.27 5.10$\pm$0.81 5.55$\pm$1.31

[Figure]

3.00±0.58 Blanco et al. 1.43±0.24 1.64±0.27 isoprene N2 760 1.63±0.15 1.61±0.14 5.84±0.80 1.62±0.11 1.60±0.13 1-pentene N2 760 1.12±0.12 1.21±0.19 5.71±0.62 1.09±0.13 1.33±0.10 1.33±0.10 isoprene N2 – O2 (20) 760 1.57±0.14 1.59±0.11 5.74±0.18 5.74±0.18 1.61±0.11 isoprene N2 600 1.58±0.09 1.55±0.10 5.61±0.25 5.61±0.25 1.53±0.13 isoprene N2 500 1.59±0.11 1.57±0.10 5.67±0.18 5.67±0.18 1.55±0.12 310±2 1,3-butadiene N2 760 1.49±0.12 1.50±0.13 4.67±0.06 4.20±0.47 1.51±0.16 isoprene N2 760 1.24±0.10 1.29±0.13 4.13±0.22 1.37±0.10 1.26±0.09 1-pentene N2 760 0.93±0.01 0.89±0.07 3.82±0.38 0.86±0.06 330±2 1,3-butadiene N2 760 1.35±0.14 1.33±0.13 4.06±0.15 3.45±0.25 1.31±0.11 isoprene N2 760 1.24±0.10 1.15±0.13 3.36±0.20 1.13±0.09 1.09±0.08 1-pentene N2 760 0.79±0.04 0.80±0.07 2.95±0.07 0.81±0.04 350±2 1,3-butadiene N2 760 1.29±0.15 1.30±0.14 3.43±0.10 2. 89±0.28 1.32±0.12 isoprene N2 760 0.98±0.07 1.06±0.13 2.84±0.26 1.19±0.10 1.01±0.08 1-pentene N2 760 0.74±0.05 0.74±0.10 2.42±0.03 0.75±0.06 363±2 1,3-butadiene N2 760 1.24±0.09 1.26±0.11 3.31±0.13 2.60±0.19 1.28±0.12 isoprene N2 760 0.87±0.4 0.86±0.08 3.01±0.10 0.88±0.2 0.85±0.6 1-pentene N2 760 0.74±0.02 0.72±0.11 2.18±0.09 0.72±0.06 0.71±0.05 1,3-butadiene N2 – O2 (20) 760 1.30±0.11 1.28±0.11 3.38±0.10 3.38±0.10 1.27±0.09

Table 2: Relative rate measurements for the reaction of Cl atoms with 5-hexen-2-one over the temperature range of 298-363K relative to 1,3-butadiene, isoprene and 1-pentene. T (K) Reference compound Bath gas (Torr of O2) Pressure in Torr (ksample/ kreference)±2$\sigma$ (ksample/ kreference)Average±2$\sigma$ (k±2$\sigma$)×10-10 (cm3molecule-1s-1) (kAverage±2$\sigma$)×10-10 (cm3molecule-1s-1) Lite.k×10-10 (cm3molecule-1s-1) at 298K 298±2 1,3-butadiene N2 760 1.24±0.13 1.26±0.15 4.19±0.63 4.14±1.25 3.15±0.5 Blanco et al. 1.26±0.10 1.29±0.11 isoprene N2 760 1.18±0.09 1.09±0.20 3.95±0.97 1.14±0.10 0.97±0.16 1-pentene N2 760 0.87±0.09 0.91±0.16 4.27±0.38 0.95±0.08 0.91±0.08 isoprene N2 – O2 (20) 760 1.13 ±0.11 1.12±0.12 4.03±0.56 4.03±0.56 1.11±0.09 isoprene N2 600 1.15±0.12 1.17±0.13 4.21±0.61 4.21±0.61 1.19±0.10 isoprene N2 500 1.16±0.14 1.15±0.15 4.16±0.58 4.16±0.58 1.15±0.11 310±2 1,3-butadiene N2 760 1.14±0.12 1.13±0.13 3.51±0.06 3.68±0.30 1.12±0.10

isoprene N2 760 1.06±0.08 1.09±0.09 3.50±0.28 1.13±0.06 1-pentene N2 760 0.94±0.08 0.94±0.12 4.04±0.04 0.95±0.09 330±2 1,3-butadiene N2 760 1.11±0.13 1.10±0.16 3.37±0.03 3.20±0.15 1.10±0.12 isoprene N2 760 1.05±0.06 1.07±0.09 3.11±0.14 1.09±0.08 1-pentene N2 760 0.84±0.06 0.84±0.08 3.12±0.03 0.85±0.04 350±2 1,3-butadiene N2 760 1.02±0.14 1.04±0.16 2.73±0.13 2.91±0.25 1.06±0.09 isoprene N2 760 1.17±0.16 1.14±0.17 3.07±0.18 1.12±0.13 1-pentene N2 760 0.92±0.10 0.90±0.14 2.94±0.13 0.89±0.09 363±2 1,3-butadiene N2 760 1.05±0.12 1.08±0.17 2.86±0.18 2.70±0.31 1.09±0.09 1.12±0.11 isoprene N2 760 0.98±0.07 1.02±0.13 2.60±0.25 0.99±0.09 1.10±0.08 1-pentene N2 760 0.89±0.05 0.88±0.08 2.65±0.03 0.87±0.06 1,3-butadiene N2 – O2 (20) 760 1.11±0.09 1.09±0.10 2.88±0.10 2.88±0.10 1.08±0.07

R3Q13. Authors report uncertainties in the abstract and in all other sections of the paper, tables etc., but there is no explanation of how were they determined and what they represent. Do they include systematic uncertainties and if they do, how were they estimated? An error analysis section will assist the reader to go through the paper and assess the reliability of the measurements. Response: We have re calculated errors and a separate section was included on error analysis in the RMS as mentioned below. "The uncertainties in the temperature (within ±2K) and pressure (within ±1 Torr) in the reaction chamber were very small and did not contribute significantly on the determination of the rate coefficients. The elution of the test molecules and reference compounds in the GC are precise and the uncertainty in concentrations was estimated to be less than 5%. For each experiment, the obtained slopes (using linear least squares method) along with the errors (95% confidence limit) are given in Tables 1 and 2. The uncertainties on the weighted average slopes ((ksample/kreference)Average) are determined using the error propagation method according to the equation: $\Delta y/y = [[\Delta a/a]2+[\Delta b/b]2+.....]1/2$, where $\Delta y/y$ is the relative error on the average slope and [$\Delta a/a$], [$\Delta b/b$] are the relative errors on the individual slopes. The errors quoted for the rate coefficients also include the quoted error in the rate coefficients for the reference reactions and are calculated using the standard error propagation method which

was used by several groups (Blanco et al., 2009; Stoeffler et al., 2013; Peirone et al., 2014 and Dash et al., 2015 ) according to the equation: $\Delta k_{test} = k_{test} \times [(\Delta k_{ref}/k_{ref})^2 + (\Delta(k_{test}/k_{ref})/(k_{test}/k_{ref}))^2]^{1/2}$, where $(\Delta k_{ref}/k_{ref})$ and $\Delta(k_{test}/k_{ref})/(k_{test}/k_{ref})$ are the relative errors on $k_{ref}$ and $k_{test}/k_{ref}$, respectively. At every temperature, the uncertainties in the averaged rate coefficients were calculated according to the equation: $\Delta k_{average} = k_{average} \times [[\Delta l/k_l]^2 + [\Delta m/k_m]^2 + [\Delta n/k_n]^2]^{1/2}$, where $\Delta l$, $\Delta m$ and $\Delta n$, are the relative errors on the individual rate coefficients and $k_l$, $k_m$ and $k_n$ are individual rate coefficients. A major source of systematic errors in the determination of the title reaction's rate coefficients are from the absolute uncertainties in the rate coefficients of the reference reactions. Blanco, M. B., Bejan, I., Barnes, I., Wiesen, P., Teruel, M. A. Temperature-dependent rate coefficients for the reactions of Cl atoms with methyl methacrylate, methyl acrylate and butyl methacrylate at atmospheric pressure. Atmos. Environ. 43, 5996–6002, 2009. Stoeffler, C., Joly, L., Durry, G., Cousin, J., Dumelie, N., Bruyant, A., Roth, E., Chakir, A. Kinetic study of the reaction of chlorine atoms with hydroxyacetone in gas-phase. Chem. Phys. Lett. 590, 221–226, 2013. Peirone, S. A., Barrera, J. A., Taccone, R. A., Cometto, P. M., Lane, S. I. Relative rate coefficient measurements of OH radical reactions with (Z)-2-hexen-1-ol and (E)-3-hexen-1-ol under simulated atmospheric conditions. Atmos. Environ. 85, 92-98, 2014. Dash, M. R., Srinivasulu, G., Rajakumar, B. Experimental and computational investigation on the gas phase reaction of p-cymene with Cl atoms. J. Phys. Chem. A 119, 559−570, 2015.

R3Q14. In general, the present reviewer believes that the submitted draft needs severe reorganization as theoretical calculations and experimental results for each ketone have substantial similarities. It would be more meaningful to present experimental results for both ketones and they should be separate from the corresponding theoretical ones, to avoid repetitions. Then, the authors might include a section where all the important observations would be summarized and comparisons between experiments and theory, relative positioning of the double bond and carbonyl group (experiment vs experiment and theory vs theory) should be highlighted and interpreted in terms of

reactivity. Response: We have reorganized the paper and all experimental and all theoretical results in separate sections for better comparison between similar molecules.

R3Q15. Although CH3CH=CHC(O)CH3 vapour pressure is low, it should be adequate for relative rate measurements, especially at higher temperatures. Experiments, even at those conditions would assist to understand how the relative positioning of the double bond and carbonyl group affects unsaturated ketones reactivity, as well as what the impact to the compete reaction mechanisms of abstraction and association is. It might also be of worth to try to do some experiments with the title ketone, especially at high temperatures. Response: In the present manuscript, the experiments were performed on 4-hexen-3-one and 5-hexen-2-one only. Just to understand the mechanism for the reactions of Cl atom reactions with unsaturated ketones, 3-penten-2-one was also studied using computational methods along with 4-hexen-3-one and 5-hexen-2-one.

R3Q16. Regarding theoretical calculations: a. How the authors verified that they have located global minima on the PES, during geometries optimization. b. Although the authors employed a level of theory of high accuracy (CCSD(T)), the basis set used was unexpectedly limited. Why did the authors chose a non-correlation consistent, double zeta basis set for single point calculations? c. Although it is reasonable, the authors should clearly state that they have also calculated ZPE using MP2 method and that they have calculated their values at 296 K. Response: We have optimized all the possible conformers during geometry optimization. The lowest energy conformers were considered for rate coefficients calculations. Whereas other possible conformers are more than 1.9 kcal mol-1 higher in energy than the lowest energy conformers and therefore, it is unlikely to have significant contribution to the reaction in the temperature range of our study. We employed 6-311++G(d,p), cc-pvdz and aug-cc-pvdz basis sets for single point energy calculations of the title reactions. The obtained rate coefficients with these basis sets were over estimated when compared with the present experimental and reported rate coefficients for the title reactions. Therefore, 6-31+G(d,p) basis

set was used for single point calculations and ZPE energies obtained at MP2 level of theory were also included in our calculations.

R3Q17. Reference reactions were measured in a narrower temperature range compared to the studied reactions. This is to say that the authors have assumed that reaction mechanism does not change as a function of temperature. This might be partly true and safe when reactions proceed only via –H abstraction, which is not the case here, where abstraction and association compete to each other and their contribution is expected to vary with temperature. It is strongly recommended the authors to avoid such assumptions and hypothetical extrapolations. It would have also been more appropriate to use as references, reactions that the rate coefficient is not pressure dependent. On the top of that, the authors recognize that pressure dependent measurements would assist to understand the mechanism of the reaction and they will reveal how the competition between the pressure independent channels of H abstraction competes with the pressure dependent pathway of Cl association to the double bond. From an atmospheric perspective, those very short lived compounds (VSLC) will not be substantially transferred to the upper troposphere ad thus rate coefficient measurements at atmospheric pressure are the most relevant ones with regard to the atmospheric impact. Response: Several groups have measured the rate coefficients of Cl atom reactions with many unsaturated hydrocarbons at room temperature (298K). However, temperature dependent rate coefficients are available only for 1-pentene (Coquet et al., 2000) and isoprene (Bedjanian et al., 1998) in the temperature range of 233-320K. Also, the rate coefficients of these reactions are close to the rate coefficients of the title reactions, which is a pre requisite to use them as reference compounds in the relative rate method. Therefore, 1-pentene and isoprene were used as reference compounds in the present investigation. As reviewer rightly pointed out, our present studies were carried out in the temperature range of 298-363K whereas, the reference reaction's rate coefficients are available in the temperature range of 233-320K only. Therefore, technically the measured rate coefficients in the higher temperature range (321-363K) may not be reliable. Recently, we have measured the temperature dependent rate

coefficients in the temperature range of 269-363K for the reaction of Cl atoms with 1,3-butadiene (Vijayakumar et al., 2017). Now, we have measured the temperature dependent rate coefficients (in the temperature range of 298-363K) for the title reactions using 1,3-butadiene + Cl reaction as a third reference reaction . The measured rate coefficients are given in Tables 1 and 2. From these tables, it is clear that the rate coefficients obtained using all the three reference compounds (1,3-butadiene, isoprene and 1-pentene) are very close to each other over the studied temperature range within the experimental uncertainties. Therefore, the obtained rate coefficients relative to 1,3-butadiene, isoprene and 1-pentene were averaged at the respective temperatures. With this additional input, the rate coefficient data obtained is reliable in the entire studied range of temperature. Vijayakumar, S., Rajakumar, B. Experimental and theoretical investigations on the reaction of 1,3-butadiene with Cl atom in the gas phase. J. Phys. Chem. A 121, 1976-1984, 2017. R3Q18. Mechanism was not thoroughly investigated and no critical SOA formation intermediates were identified. The part that the authors try to interpret the mechanism in terms of electronic effects due to the presence of carbonyl group is incomplete and steric factors and entropy change to the TS should be also considered. A comparative plot of the abstraction and association kinetics as a function of temperature would assist to elucidate the relative importance. Response: As we have discussed in section 3.7, the atmospheric degradation mechanism was proposed till the chloroalkyl radicals are formed. It was proposed based on our theoretical observations. The contribution of abstraction channels as a function of temperature is given in Figures 12 to 14 (branching ratios plots) and the same discussed in the text (c.f. section 3.4). In fact the branching ratios clearly reveal that the association reactions are dominant. The variation of the rate coefficients with respect to temperature both in case of abstraction and association is shown in the branching ratio plots. Tables: R3Q19. Relative rate tables: In all cases ratios uncertainties and bimolecular rate coefficients ones are identical, which is to say that the authors have not propagated the error limits. Revision is required through the whole draft. (e.g., Table 1. r = 1.46 $\pm$0.36 and k =(5.08$\pm$0.36) $\times$10-10 cm3 molecule-1 s-1). The sources

of the quoted uncertainties should also be included. Response: Complete error analyses was carried out again and now we have reported the corrected errors. For details, please refer to our response to R3Q13 of this reviewer.

R3Q20. Table 4. The authors report a rate coefficient of 1.09 ×10-9 cm3 molecule-1 s-1 at 275K that is higher than the collision rate limit (∼8 ×10-10 cm3 molecule-1 s-1 at room temperature). This is possible only when reaction follows a Harpoon mechanism, which means ion-ion or ion-molecule interaction, which definitely is not the case here. Most likely, this is due to intrinsic problems of the computational methods and if there was an estimated uncertainty of the calculated k values via benchmarking calculations, the error limits would have revealed that this is due to the limitations of the theoretical methods employed. Response: In case of Cl atom reaction with 5-hexen-2-one (c.f. Table 4), the rate coefficients for the reaction below 275 K are touching the gas kinetic limit. This kind of behavior is expected in case of the ion-ion or ion-molecule interaction or radical – radical reactions. May be this is due to the presence of Cl atom/radical in the title reaction. The reaction is totally governed by the addition of Cl atom at the double bond present in the substrate. Addition channels are exothermic. Also, these channels are feasible both kinetically (because of the submerged transition states) and thermodynamically. As the reaction is dominant and proceed via negative transition states, the rate of reaction increases with the decrease in temperature. All these would probably favor the reaction at low temperatures to the largest possible extent. In addition, as the reviewer rightly pointed, there may be intrinsic limitations of the theory chosen for these calculations as well. This is added in the RMS.

Graphs: R3Q21. All the relative rate plots should begin from zero difference in test and reference concentration relative variance (0,0). Negative values are meaningless. Moreover, figure captions should include all the experimental conditions and explain everything that is shown inside the plots and in insets of the graph. Finally, for the shake of clarity, please use different symbols when different references are included in the graph (Not only different colors). Response: It was corrected in the RMS as shown

[Figure]

below.

Figure 1: Plot of the relative decrease in the concentration of 4-hexen-3-one due to its reaction with Cl atoms relative to 1,3-butadiene, isoprene and 1-pentene at 298K and 760 Torr of N2. The symbols indicate measurements made relative to different compounds as indicated in the legend. These lines are linear least squares fits of the data to equation 1 that yield rate coefficients for reaction 1. The error bars are from the precision of the measurement.

Figure 3: Plot of the relative decrease in the concentration of 5-hexen-2-one due to its reaction with Cl atoms relative to 1,3-butadiene, isoprene and 1-pentene at 298K and 760 Torr of N2. The symbols indicate measurements made relative to different compounds as indicated in the legend. These lines are linear least squares fits of the data to equation 1 that yield rate coefficients for reaction 2. The error bars are from the precision of the measurement.

R3Q22. In Arrhenius plots, first, include an X-mirror axis so as the reader to have a measure of temperature. Also, use k (and log axis) in Y-axis. Note that lnk has no units and if the authors decide to keep it this way, they should refer to it as ln (k, cm3 molecule-1 s-1). Response: It was corrected in the RMS as shown below.

Figure 2: Arrhenius plot of CVT/SCT rate coefficients calculated at the CCSD(T)/6-31+G(d, p)//MP2/6-311++G(d,p) level of theory between the temperatures 275 and 400 K and experimentally measured rate coefficients between the temperatures of 298 and 363 K for the reaction of Cl atoms with 4-hexen-3-one.

Figure 4: Arrhenius plot of CVT/SCT rate coefficients obtained at the CCSD(T)/6-31+G(d, p)//MP2/6-311++G(d,p) level of theory between the temperatures 275 and 400 K and experimentally measured rate coefficients between the temperatures of 298 and 363 K for the reaction of Cl atoms with 5-hexen-2-one.

Figure 11: Arrhenius plot of CVT/SCT rate coefficients obtained at the CCSD(T)/6-

31+G(d, p)// MP2/6-311++G(d,p) level of theory between the temperatures 275 and 400 K for the reaction of Cl atoms with 3-penten-2-one. R3Q23. In some cases, curvature was observed in Arrhenius plot. Why didn't the authors fit their data using a modified version of Arrhenius fitting (A ×Tn), which include temperature dependence on the pre-exponential factor and is quite common in association reactions. Response: In Figure 2, after adding third reference compound, it is showing a liner behavior and used linear least square method to fit the experimental data and shown below.

Figure 2: Arrhenius plot of CVT/SCT rate coefficients calculated at the CCSD(T)/6-31+G(d, p)//MP2/6-311++G(d,p) level of theory between the temperatures 275 and 400 K and experimentally measured rate coefficients between the temperatures of 298 and 363 K for the reaction of Cl atoms with 4-hexen-3-one. Conclusions: R3Q24. Conclusions are very limited and not appropriate for such an amount of work. Response: We have included some more points in the RMS as mentioned below. "In this study, reactions of Cl atoms with 4-hexen-3-one (R1), 5-hexen-2-one (R2) and 3-penten-2-one (R3) were investigated. R1 and R2 were investigated experimentally in the temperature range of 298-363K; whereas R1, R2 and R3 were studied computationally. The rate coefficients for all the reactions were calculated by CVT/SCT coupled with CCSD(T)/6-31+G(d,p)//MP2/6-311++G(d,p) level of theory. Addition of Cl atom across the double bond in all the reactions predominates and abstraction of hydrogen would have very insignificant contribution to the overall reaction. From both experimental and theoretical measurements, negative temperature dependence was observed over the studied temperature range because of the submerged transition states for addition channels. Thermodynamically, the addition reactions are more feasible and spontaneous. The reactions of Cl atoms with test molecules are much faster than the other dominant reactions (OH radicals, O3 molecules and NO3 radicals) especially in polluted mid –continental regions, in industrial locations, in marine boundary layers and in urban polluted areas where the Cl atom concentrations reaches maximum up to 105 atoms cm−3. The cumulative lifetimes of the test molecules are very low and they are lost within few hours as soon as they are released into the atmosphere. The Cl atom initiated reactions with unsaturated ketones leads to the formation of halogenated ketones and a variety of organic nitrates, in the nitrogen rich environment. On further reactions with NOx, these compounds form thermally stable secondary organic nitrates which may show significant impact on the air quality and climate change of the Earth's atmosphere. However, the atmospheric lifetimes of unsaturated ketones are relatively short which suggest their inconsiderable impact on the global warming. The degradation of unsaturated ketones would lead to significant amount of ozone formation in the troposphere". This is added in the RMS.

R3Q25. GWP contribution is negligible, since the studied ketones contain no C-F bonds and more importantly they are very short lived compounds. Did the authors look if there are any available IR spectra in the literature that they can comment on them to justify their statements? Response: We have calculated GWPs for test molecules and estimated them to be 0.01, 0.01 and 0.01 for 4-hexen-3-one, 5-hexen-2-one and 3-penten-2-one respectively for 20 years time horizon. Hence, we have stated that GWPs for test molecules seems to be negligible. We have searched for the IR spectra in the literature and are not reported till date to the best of our knowledge. Therefore, we are not in a position to compare our data.

R3Q26. The authors could estimate a POCP (photochemical ozone creation potentials) for the studied ketones and compare with similar unsaturated compounds. This is expected to be the major contribution of those compounds on air-quality issues. Response: We have calculated ozone formation potentials in the troposphere for the title reactions and found to be 7, 6 and 5 ppm for 4-hexen-3-one, 5-hexen-2-one and 3-penten-2-one respectively. The degradation of unsaturated ketones would lead to significant amount of ozone formation in the troposphere. This is added in the RMS.

R3Q27. Cl chemistry importance should be commented in conclusion part. Although it is not expected to substantially affect the fate of ketones, since they are extremely short lived compounds, it might be of importance if chlorinated products are formed. In particular for such short lived compounds it might also affect their POCP, taking

into account that metropolitan cities and polluted areas are responsible for the huge majority of emissions of such compounds and in most of the cases they are located near by the sea, where the higher levels of Cl atoms are observed. Response: The importance of the Cl atom chemistry is discussed for the query 1 (R3Q24) of the conclusion part. This is added in the RMS.

Please also note the supplement to this comment:
https://www.atmos-chem-phys-discuss.net/acp-2017-163/acp-2017-163-AC3-supplement.pdf

---

## Author Comment (AC2)

Review of Cl + alkenenones for ACPD by Theodore S. Dibble (tsdibble@esf.edu)

We thank the reviewer for going through the manuscript entitled **"Experimental and computational kinetics investigations for the reactions of Cl atoms with unsaturated ketones in gas phase"** and for his/her constructive suggestions to improve the quality of the manuscript. We have incorporated all the suggestions and given explanations to the queries in the revised manuscript (RMS) at appropriate places. The changes/additions in the revised manuscript are given in blue color for ready reference. The complete rebuttal is given below.

GENERAL COMMENTS

R2Q1. This manuscript reports relative rate experiments and computations on the kinetics of Cl reacting with a series of unsaturated ketones as a function of temperature. These compounds are of some interest and the reaction of Cl with these compounds can contribute significantly to their loss under some atmospheric conditions. The experiments (on two compounds) appear to be solid, although additional details should be reported and the clarity improved. Also, there is no experimental rate constant data on the reference compounds at the higher temperatures studied here. As a result, the absolute rate constants (as opposed the relative rate constants) at those higher temperatures may not be reliable.

**Response:** Several groups have measured the rate coefficients of Cl atom reactions with many unsaturated hydrocarbons at room temperature (298K). However, temperature dependent rate coefficients are available only for 1-pentene (Coquet et al., 2000) and isoprene (Bedjanian et al., 1998) in the temperature range of 233-320K. Also, the rate coefficients of these reactions are close to the rate coefficients of the title reactions, which is a pre requisite to use them as reference compounds in the relative rate method. Therefore, 1-pentene and isoprene were used as reference compounds in the present investigation. As reviewer rightly pointed out, our present studies were carried out in the temperature range of 298-363K whereas, the reference reaction's rate coefficients are available in the temperature range of 233-320K only. Therefore, technically the measured rate coefficients in the higher temperature range (321-363K) may not be reliable. Recently, we have measured the temperature dependent rate coefficients in the temperature range of 269-363K for the reaction of Cl atoms with 1,3-butadiene (Vijayakumar et al., 2017). Now, we have measured the temperature dependent rate coefficients (in the temperature range of 298-

363K) for the title reactions using 1,3-butadiene + Cl reaction as a third reference reaction . The measured rate coefficients are given in Tables 1 and 2. From these tables, it is clear that the rate coefficients obtained using all the three reference compounds (1,3-butadiene, isoprene and 1-pentene) are very close to each other over the studied temperature range within the experimental uncertainties. Therefore, the obtained rate coefficients relative to 1,3-butadiene, isoprene and 1-pentene were averaged at the respective temperatures. With this additional input, the rate coefficient data obtained is reliable in the entire studied range of temperature.

Vijayakumar, S., Rajakumar, B. Experimental and theoretical investigations on the reaction of 1,3-butadiene with Cl atom in the gas phase. J. Phys. Chem. A 121, 1976-1984, 2017.

**Table 1**: Relative rate measurements for the reaction of Cl atoms with 4-hexen-3-one over the temperature range of 298-363K at 760 Torr in $N_2$ relative to 1,3-butadiene, isoprene and 1-pentene.

| T(K) | Reference compound | Bath gas (Torr of $O_2$) | Pressure in Torr | $(k_{sample}/k_{reference})$ $\pm 2\sigma$ | $(k_{sample}/k_{reference})$ Average $\pm 2\sigma$ | $(k\pm2\sigma)\times10^{-10}$ $cm^3 molecule^{-1}s^{-1}$ | $(k\pm2\sigma)\times10^{-10}$ $cm^3 molecule^{-1}s^{-1}$ | Lit. $(k\pm2\sigma)\times10^{-10}$ $(cm^3 molecule^{-1}s^{-1})$ at 298K |
|---|---|---|---|---|---|---|---|---|
| 298±2 | 1,3-butadiene | $N_2$ | 760 | 1.55±0.21
1.43±0.24
1.64±0.27 | 1.54±0.27 | 5.10±0.81 | | |
| | isoprene | $N_2$ | 760 | 1.63±0.15
1.62±0.11
1.60±0.13 | 1.61±0.14 | 5.84±0.80 | 5.55±1.31 | |
| | 1-pentene | $N_2$ | 760 | 1.12±0.12
1.09±0.13
1.33±0.10
1.33±0.10 | 1.21±0.19 | 5.71±0.62 | | |
| | isoprene | $N_2 - O_2$ (20) | 760 | 1.57±0.14
1.61±0.11 | 1.59±0.11 | 5.74±0.18 | 5.74±0.18 | 3.00±0.58 Blanco et al. |
| | isoprene | $N_2$ | 600 | 1.58±0.09
1.53±0.13 | 1.55±0.10 | 5.61±0.25 | 5.61±0.25 | |
| | isoprene | $N_2$ | 500 | 1.59±0.11
1.55±0.12 | 1.57±0.10 | 5.67±0.18 | 5.67±0.18 | |
| 310±2 | 1,3-butadiene | $N_2$ | 760 | 1.49±0.12
1.51±0.16 | 1.50±0.13 | 4.67±0.06 | | |
| | isoprene | $N_2$ | 760 | 1.24±0.10
1.37±0.10
1.26±0.09 | 1.29±0.13 | 4.13±0.22 | 4.20±0.47 | |
| | 1-pentene | $N_2$ | 760 | 0.93±0.01 | 0.89±0.07 | 3.82±0.38 | | |

| T (K) | Reference compound | Bath gas (Torr of $O_2$) | Pressure in Torr | $(k_{sample}/k_{reference})\pm 2\sigma$ | $(k_{sample}/k_{reference})_{Average}\pm 2\sigma$ | $(k\pm 2\sigma)\times 10^{-10}$ | $(k_{Average}\pm 2\sigma)\times 10^{-10}$ | Lit.k$\times 10^{-10}$ |
|---|---|---|---|---|---|---|---|---|
| | | | | 0.86±0.06 | | | | |
| 330±2 | 1,3-butadiene | $N_2$ | 760 | 1.35±0.14
 1.31±0.11 | 1.33±0.13 | 4.06±0.15 | 3.45±0.25 | |
| | isoprene | $N_2$ | 760 | 1.24±0.10
 1.13±0.09
 1.09±0.08 | 1.15±0.13 | 3.36±0.20 | | |
| | 1-pentene | $N_2$ | 760 | 0.79±0.04
 0.81±0.04 | 0.80±0.07 | 2.95±0.07 | | |
| 350±2 | 1,3-butadiene | $N_2$ | 760 | 1.29±0.15
 1.32±0.12 | 1.30±0.14 | 3.43±0.10 | 2. 89±0.28 | |
| | isoprene | $N_2$ | 760 | 0.98±0.07
 1.19±0.10
 1.01±0.08 | 1.06±0.13 | 2.84±0.26 | | |
| | 1-pentene | $N_2$ | 760 | 0.74±0.05
 0.75±0.06 | 0.74±0.10 | 2.42±0.03 | | |
| 363±2 | 1,3-butadiene | $N_2$ | 760 | 1.24±0.09
 1.28±0.12 | 1.26±0.11 | 3.31±0.13 | 2.60±0.19 | |
| | isoprene | $N_2$ | 760 | 0.87±0.04
 0.88±0.02
 0.85±0.06 | 0.86±0.08 | 3.01±0.10 | | |
| | 1-pentene | $N_2$ | 760 | 0.74±0.02
 0.72±0.06
 0.71±0.05 | 0.72±0.11 | 2.18±0.09 | | |
| | 1,3-butadiene | $N_2 - O_2$ (20) | 760 | 1.30±0.11
 1.27±0.09 | 1.28±0.11 | 3.38±0.10 | 3.38±0.10 | |

**Table 2**: Relative rate measurements for the reaction of Cl atoms with 5-hexen-2-one over the temperature range of 298-363K with reference to 1,3-butadiene, isoprene and 1-pentene.

| T (K) | Reference compound | Bath gas (Torr of $O_2$) | Pressure in Torr | $(k_{sample}/k_{reference})\pm 2\sigma$ | $(k_{sample}/k_{reference})_{Average}\pm 2\sigma$ | $(k\pm 2\sigma)\times 10^{-10}$ (cm$^3$molecule$^{-1}$s$^{-1}$) | $(k_{Average}\pm 2\sigma)\times 10^{-10}$ (cm$^3$molecule$^{-1}$s$^{-1}$) | Lit.k$\times 10^{-10}$ (cm$^3$molecule$^{-1}$s$^{-1}$) at 298K |
|---|---|---|---|---|---|---|---|---|
| 298±2 | 1,3-butadiene | $N_2$ | 760 | 1.24±0.13
 1.26±0.10
 1.29±0.11 | 1.26±0.15 | 4.19±0.63 | 4.14±1.25 | 3.15±0.5
 Blanco et al. |
| | isoprene | $N_2$ | 760 | 1.18±0.09
 1.14±0.10
 0.97±0.16 | 1.09±0.20 | 3.95±0.97 | | |
| | 1-pentene | $N_2$ | 760 | 0.87±0.09
 0.95±0.08
 0.91±0.08 | 0.91±0.16 | 4.27±0.38 | | |
| | isoprene | $N_2 - O_2$ (20) | 760 | 1.13 ±0.11
 1.11±0.09 | 1.12±0.12 | 4.03±0.56 | 4.03±0.56 | |

| | | | | | | | | |
|---|---|---|---|---|---|---|---|---|
| | isoprene | $N_2$ | 600 | 1.15±0.12 / 1.19±0.10 | 1.17±0.13 | 4.21±0.61 | 4.21±0.61 | |
| | isoprene | $N_2$ | 500 | 1.16±0.14 / 1.15±0.11 | 1.15±0.15 | 4.16±0.58 | 4.16±0.58 | |
| 310±2 | 1,3-butadiene | $N_2$ | 760 | 1.14±0.12 / 1.12±0.10 | 1.13±0.13 | 3.51±0.06 | 3.68±0.30 | |
| | isoprene | $N_2$ | 760 | 1.06±0.08 / 1.13±0.06 | 1.09±0.09 | 3.50±0.28 | | |
| | 1-pentene | $N_2$ | 760 | 0.94±0.08 / 0.95±0.09 | 0.94±0.12 | 4.04±0.04 | | |
| 330±2 | 1,3-butadiene | $N_2$ | 760 | 1.11±0.13 / 1.10±0.12 | 1.10±0.16 | 3.37±0.03 | 3.20±0.15 | |
| | isoprene | $N_2$ | 760 | 1.05±0.06 / 1.09±0.08 | 1.07±0.09 | 3.11±0.14 | | |
| | 1-pentene | $N_2$ | 760 | 0.84±0.06 / 0.85±0.04 | 0.84±0.08 | 3.12±0.03 | | |
| 350±2 | 1,3-butadiene | $N_2$ | 760 | 1.02±0.14 / 1.06±0.09 | 1.04±0.16 | 2.73±0.13 | 2.91±0.25 | |
| | isoprene | $N_2$ | 760 | 1.17±0.16 / 1.12±0.13 | 1.14±0.17 | 3.07±0.18 | | |
| | 1-pentene | $N_2$ | 760 | 0.92±0.10 / 0.89±0.09 | 0.90±0.14 | 2.94±0.13 | | |
| 363±2 | 1,3-butadiene | $N_2$ | 760 | 1.05±0.12 / 1.09±0.09 / 1.12±0.11 | 1.08±0.17 | 2.86±0.18 | 2.70±0.31 | |
| | isoprene | $N_2$ | 760 | 0.98±0.07 / 0.99±0.09 / 1.10±0.08 | 1.02±0.13 | 2.60±0.25 | | |
| | 1-pentene | $N_2$ | 760 | 0.89±0.05 / 0.87±0.06 | 0.88±0.08 | 2.65±0.03 | | |
| | 1,3-butadiene | $N_2 - O_2$ (20) | 760 | 1.11±0.09 / 1.08±0.07 | 1.09±0.10 | 2.88±0.10 | 2.88±0.10 | |

R2Q2: The clarity and precision of the writing should be improved.

The computational chemistry calculations are not reliable. While it the authors could address issues with the energeties obtained from quantum chemistry, it is probably not feasible to reliably compute rate constants and branching ratios for the systems being studied here. I am uncertain as to the importance of the two compounds being studied experimentally, and the experiments only provide rate constants, not branching ratios. So the manuscript remaining after removing the computational parts probably does not belong in ACP.

**Response:** Both experimental and theoretical studies give a better understanding of the chemical reactions which are occurring in the troposphere. Experimentally, one can measure the global rate coefficient whereas theoretically, we can calculate the contribution of each reaction site towards its global rate coefficient and also gives the essential information about the reaction mechanism. In the present investigation, branching ratio calculations were performed using computationally obtained rate coefficients.

SPECIFIC COMMENTS ON EXPERIMENTS

R2Q3. Secondary chemistry appears to be a minor factor, but the reference compounds are sufficiently similar that they might give rise to similar secondary chemistry. It would help to present a brief discussion about WHY secondary chemistry is expected be minor. Just in case someone wants to model the experiment, the manuscript should include the repetition rate of the 248 nm lasers and its fluence (in mJ cm-2 pulse-1 rather than mJ pulse-1). For the same reason, the initial concentration range of the test compounds and reference compounds should be listed in the text.

**Response:** As mentioned in section 2.1, we have performed some preliminary tests before doing the experiments to check the influence of the secondary chemistry on title reactions. The reaction mixture (test molecule, reference compound and the precursor for Cl atom) was kept for 6 hours in dark which is more than the actual reaction time. The samples were analyzed in the GC at every half-an-hour and verified for any significant loss of the reactants and no such influence was observed. The sample mixture without the precursor (the test molecules and the reference compounds) was irradiated at 248 nm for 5 minutes, to verify the loss of the compounds due to direct photolysis. A maximum of 3 to 4% of the change in concentrations was observed, which indicates that neither the test molecules nor the reference compounds were dissociated by photolysis. A good way of confirming the secondary chemistry is via scavenging the radicals by adding oxygen. At room temperature and at extreme temperatures, oxygen was added to the reaction mixture and obtained rate coefficients are given in Tables 1 and 2. A maximum of 5% change was observed in the rate coefficients, which shows the negligible influence of secondary reactions due to the radicals formed in the test reaction.

The experiments were carried out with a repetition rate of 10 Hz at 248 nm wavelength and its fluence was maintained at 5-6 mJ cm$^{-2}$ pulse$^{-1}$. The typical concentrations of the reactant and

reference compounds were varied between (4-6)×10$^{16}$ molecules cm$^{-3}$ and that of oxalylchloride was maintained between (4-6)×10$^{17}$ molecules cm$^{-3}$. This is added in the RMS.

R2Q4. An expanded version of Table 1 is needed that includes the initial concentration of the reference compound. Also, results should be presented with experiments with O$_2$ present. Ideally, the Supplementary Material would include the measured concentration of the test compounds and the reference compound at 1200, 1400, 1600, 1800 and 2000 pulses for each experiment.

**Response:** The experiments which were performed in presence of O$_2$ are incorporated for the reactions of Cl atoms with 4-hexen-3-one and 5-hexen-2-one in Tables 1 and 4 respectively in the RMS. The concentrations of test molecules and the reference compounds measured using Gas Chromatography (GC) at 1200, 1400, 1600, 1800 and 2000 pulses for every experiment are given in the revised supplementary material.

R2Q5. The literature values of the rate constant for Cl + isoprene and Cl + 1-pentene are only known up to 320K. There is no reason to expect Arrhenius behavior from these reactions. Consequently, it is not appropriate to derive absolute rate constants at these temperatures without highlighting the fact that rate constants for the reference compounds are being extrapolated.

**Response:** Rate coefficients for the reference reactions (Cl + 1-pentene and isoprene) are available only up to 320K. In the lack of availability of the reference rate coefficients up to 363K, the extrapolated rate coefficients were used in the measurements. In addition, the measurements were carried out using 1,3-butadiene as a third reference. Please refer to our response to R2Q1 of this reviewer for complete description.

R2Q6. The value of [Cl] used to compute atmospheric lifetimes (in Table 10) is only valid in a small part of the atmosphere. This should be noted.

**Response:** To know the importance of the Cl atom reactions, the atmospheric lifetimes of the test molecules were estimated with respect to their reactions with Cl atoms both in ambient conditions (1.00×10$^3$ molecules cm$^{-3}$, Singh et al., 1996) and marine boundary layer (1.30×10$^5$ molecules cm$^{-3}$, Spicer et al., 1998). In ambient conditions the lifetimes of 4-hexen-3-one, 5-hexen-2-one and 3-pentene-2-one are 8, 12 and 19 days respectively. In the marine boundary

layer the lifetimes of 4-hexen-3-one, 5-hexen-2-one and 3-pentene-2-one are estimated as 3.5, 5.3 and 8.7 hours respectively. Since the Cl atom reactions are important mainly in marine boundary layer and polluted urban areas where the Cl atom concentration reaches $1.3 \times 10^5$ atoms $cm^{-3}$, it was considered when compared with other oxidants in the Table 10. This is added in the atmospheric implications (section 3.8) of the RMS.

SPECIFIC COMMENTS ON CALCULATIONS

R2Q7. The CCSD(T) energy calculations use a basis set that is far too small to be reliable. This is evident in many of the reported values of the critical energies for hydrogen abstraction (they are far too high). Could there also be a problem with unstable wavefunctions contributing to highenergies of these TSs? The small basis set could also (via basis set superposition error) lead to TS energies for the addition reaction that are lower than the actual values. The fact that the reported rate constants (dominated by addition) agree with experiment is due to fortuitous cancellation of error. A basis set extrapolation scheme or composite method is needed for accurate treatment of both types of TSs.

**Response:** We employed 6-311++G(d,p), cc-pvdz and aug-cc-pvdz basis sets for single point energy calculations of the title reactions. The obtained rate coefficients with these basis sets were over estimated when compared with the present experimental and reported rate coefficients for the title reactions. Hence, 6-31+G(d,p) basis set was used for single point calculations.

One of the composite method is G3MP2 and the obtained rate coefficients with G3MP2 are overestimated when compared with experimental and reported rate coefficients for the title reactions. However, with other level of throes and basis sets, we observed the Cl atom addition reactions are more dominant and showing negative temperature dependence over the studied temperature range for all the title reactions. We tried to optimize all the geometries with DFT and meta DFT methods such as B3LYP and M062X level of theories and were not successful in getting most of the transition states. Having this technical difficulty, the rate coefficients obtained at CCSD(T)/6-31+G(d, p)//MP2/6-311++G(d, p) level of theory were used to compared with the experimental and reported rate coefficients.

R2Q8. The addition of Cl to the test compounds to form a reactive complex must have a variational TS (in addition to the saddle point reported for the formation of a covalent chlorine

carbon bond). Strictly speaking, a 2-TS approach is needed to obtain the rate constant in the high-pressure limit (see J. Phys. Chem. A, 2006, 110, pp 6960–6970), but I would not insist on authors carrying out these calculations. The manuscript implicitly assumes the addition reactions are in the high pressure limit at 1 atm over the temperature range reported. This may be reasonable, but should be stated explicitly.

**Response:** We have checked the variational effects for all the three reactions and found that these addition channels are not having variational effects. As the reviewer rightly pointed out, we have measured the rate coefficients at the high pressure limits (1 atm. Pressure of $N_2$ and 298 to 363K). This is added in the RMS.

R2Q9. There is an issue with the computations that the authors won't be able to overcome: the general approach used here probably has limited applicability to Cl reaction with molecules containing C=C double bonds. See the 2014 paper by A.G. Suits and A. M. Mebel (DOI:10.1038/ncomms5064). This paper makes two major points relevant here:

a) H-abstraction from allylic sites proceeds without a barrier (3-penten-2-one and 4-hexen-3-one have allylic sites).

b) There exists roaming paths connecting chlorine adducts of the alkenes to HCl formation (potentially relevant to all three species studied in the manuscript under review).

The results in this paper have been verified and extended. The reaction paths described in (a) and (b) are important under the conditions of the experiment, and (a) is important. Although those conditions are far different than those in the atmosphere, another paper (DOI:10.1038/srep40105) suggests that roaming paths leading to HCl are more important at thermal energies (meaning atmospheric conditions) than the conditions of the Suits and Mebel paper.

This conclusion The results found in this manuscript are not consistent with those of the Suits and Mebel paper or subsequent computations. The roaming paths can only be treated by running dynamics, and that is far outside the scope of this manuscript. As a result, it appears that reliable kinetic insight cannot be obtained from the general approach used in the present manuscript. For those who got lost in the gory details of the physical chemistry I just discussed, just know that the theoretical approach in the present manuscript is, at least in part, inconsistent with obtaining

the ~17% branching percentage (at 298 K) found for H-abstraction in the reaction of Cl + isoprene (Bedjanian et al., 1998, cited in the manuscript).

**Response:**

a) The allylic hydrogens are TS8, TS9 and TS10; TS6 and TS7; TS6, TS7 and TS8 in 4-hexen-3-one, 5-hexen-2-one and 3-penten-2-one respectively. These hydrogen abstractions from allylic sites are thermodynamically feasible (*c.f.* Tables 6, 7 and 8) and kinetically (*c.f.* Table 11 and Figures 12, 13 and 14) not favorable with low barrier heights, which are consistent with our earlier results (Walvalkar et al., 2016 and Vijayakumar et al., 2017).

b) The Cl atom addition followed by HCl elimination (roaming path) would have effect on the determination of the total rate coefficients of the title reactions. The HCl formation in the reaction of Cl atom with isobutene via roaming mechanism was observed by Chen et al. (DOI: 10.1038/srep40105) and experimental conditions was entirely different from the present study. The roaming path approach is completely different study from the present one.

Walavalkar, M. P., Vijayakumar, S., Sharma, A., Rajakumar B., Dhanya, S. Is H atom abstraction important in the reaction of Cl with 1-alkenes? J. Phys. Chem. A 120, 4096-4107, 2016.

Vijayakumar, S., Rajakumar, B. Experimental and theoretical investigations on the reaction of 1,3-butadiene with Cl atom in the gas phase. J. Phys. Chem. A 121, 1976-1984, 2017.

R2Q10. As Anonymous Referee #1 points out, hindered rotor corrections to rate constants may be important.

**Response:** Hindered rotor (HR) calculations were performed and compared with the present experimental and reported rate coefficients (given below). Rate coefficients obtained including HR corrections are almost equal to our earlier theoretical calculations. As the reviewer rightly pointed, the discrepancy between theoretical and the present experimental rate coefficients may be due to the errors in pre-exponential factors and the errors in the estimation of barrier heights. As the rate coefficient at a given temperature is the combination of both pre-exponential factor and the activation energy, the difference can be attributed to the accuracy with which both these factors are determined. The pre-exponential factor depends on how best the partition functions of reactants and transition states are estimated, which in turn depends on the vibrational frequencies

obtained in the calculations. On another hand, the uncertainties in the calculated energies of transition states can critically affect the calculated rate coefficients. Lynch *et al*., 2001;Ali *et al*., 2016; and Ali *et al*., 2015 concluded that, there would be an error of about 1.1 kcal mol$^{-1}$ in the barrier height calculations at the CCSD(T) level of theory with 6-31+G(d,p) basis set. The same level of theory and the basis set were used in the present calculations. Therefore, given an uncertainty of about 1 kcal mol$^{-1}$ in the activation barrier, the theoretically calculated rate coefficients are in reasonable agreement with the reported experimentally measured ones. This discussion is added in the RMS.

**Table:** Comparison of the rate coefficients (cm$^3$ molecule$^{-1}$ s$^{-1}$) for the reactions of unsaturated ketones with Cl atoms at 298K.

|  | **4-hexen-3-one + Cl** | **5-hexen-2-one + Cl** | **3-penten-2-one + Cl** |
|---|---|---|---|
| k $_{Theory}$ | $3.66 \times 10^{-10}$ | $5.56 \times 10^{-10}$ | $2.4 \times 10^{-10}$ |
| k $_{Theory\ with\ HRcorrection}$ | $3.60 \times 10^{-10}$ | $5.47 \times 10^{-10}$ | $2.38 \times 10^{-10}$ |
| k $_{Experimental}$ | $(5.55 \pm 1.31) \times 10^{-10}$ | $(4.14 \pm 1.25) \times 10^{-10}$ | - |
| k $_{Blanco\ et\ al.}$ | $(3.00 \pm 0.58) \times 10^{-10}$ | $(3.15 \pm 0.50) \times 10^{-10}$ | $(2.53 \pm 0.54) \times 10^{-10}$ |

**Table**: Comparison of the theoretically obtained rate coefficients (cm$^3$ molecule$^{-1}$ s$^{-1}$) for the reaction of Cl atoms with unsaturated ketones at CCSD(T)/6-31+G(d, p)//MP2/6-311++G (d, p) level of theory over the temperature range of 275-400K.

| T (K) | 4-hexen-3-one + Cl | | 5-hexen-2-one + Cl | | 3-penten-2-one + Cl | |
|---|---|---|---|---|---|---|
|  | k $_{Theory}$ | k $_{Theory\ with\ HR\ correction}$ | k $_{Theory}$ | k $_{Theory\ with\ HR\ correction}$ | k $_{Theory}$ | k $_{Theory\ with\ HR\ correction}$ |
| 275 | $5.81 \times 10^{-10}$ | $5.73 \times 10^{-10}$ | $1.09 \times 10^{-09}$ | $1.07 \times 10^{-09}$ | $3.51 \times 10^{-10}$ | $3.48 \times 10^{-10}$ |
| 298 | $3.66 \times 10^{-10}$ | $3.60 \times 10^{-10}$ | $5.56 \times 10^{-10}$ | $5.47 \times 10^{-10}$ | $2.40 \times 10^{-10}$ | $2.38 \times 10^{-10}$ |
| 325 | $2.33 \times 10^{-10}$ | $2.29 \times 10^{-10}$ | $3.85 \times 10^{-10}$ | $3.78 \times 10^{-10}$ | $1.66 \times 10^{-10}$ | $1.65 \times 10^{-10}$ |
| 350 | $1.64 \times 10^{-10}$ | $1.62 \times 10^{-10}$ | $2.44 \times 10^{-10}$ | $2.40 \times 10^{-10}$ | $1.26 \times 10^{-10}$ | $1.24 \times 10^{-10}$ |
| 375 | $1.22 \times 10^{-10}$ | $1.20 \times 10^{-10}$ | $1.57 \times 10^{-10}$ | $1.54 \times 10^{-10}$ | $9.89 \times 10^{-11}$ | $9.79 \times 10^{-11}$ |
| 400 | $9.49 \times 10^{-11}$ | $9.35 \times 10^{-11}$ | $1.00 \times 10^{-10}$ | $9.83 \times 10^{-11}$ | $8.07 \times 10^{-11}$ | $7.99 \times 10^{-11}$ |

Lynch, B. J. and Truhlar, D. G.: How well can hybrid density functional methods predict transition state geometries and barrier heights? J. Phys. Chem. A 105, 2936-2941, 2001.

Ali, M. A., Sonk, J. A. and Barker, J. R.: Predicted chemical activation rate constants for $HO_2$ + $CH_2NH$: The dominant role of a hydrogen bonded pre-reactive complex. J. Phys. Chem. A 120, 7060-7070, 2016.

Ali, M.A. and Barker, J.R.: Comparison of there isoelectronic multiple-well reaction systems: $OH+CH_2O$, $OH+CH_2CH_2$, and $OH+CH_2NH$. J. Phys. Chem. A 119, 7578-7592, 2015.

TECHNICAL CORRECTIONS

R2Q11. The GC temperature program and flow rate should be specified (at least in the Supplementary Material).

**Response:** The following conditions were maintained during the GC analyses of the reaction mixtures.

Inlet temperature: $160^oC$

Pressure: 24.05 PSI

HP Plot Q Column flow: 1.96 ml min$^{-1}$

Pressure: 24.05 PSI

Oven temperature: $220^oC$

Run time: 6 minutes

FID detector temperature: $240^oC$

The above information is given in the revised supplementary material.

R2Q12. On page 5 where the absence of loss of test and reference compounds was verified in the dark and in the absence of oxalyl chloride, please specify the upper limit to the loss (e.g., < 4%). Similarly, specify the upper limit to the change in rate constant upon adding $O_2$ (and the partial pressure of $O_2$ used).

**Response:** The sample mixture without the precursor was irradiated at 248 nm for 5 minutes, to verify the loss of the compounds due to direct photolysis and a maximum of 3 to 4% of the change in concentrations was observed which indicates non-interference of the secondary chemistry on the title reactions. About 20 Torr of oxygen was added to the reaction mixture (maintained at 760 Torr) and the experiments were carried out at room temperature and at

extreme temperatures. The obtained rate coefficients are given in Tables 1 and 2. Maximum 5% change was observed in the rate coefficients which shows the negligible influence of secondary reactions due to the radicals formed in the test reaction. This is added in the RMS.

R2Q13. The sample of 4-hexen-3-one is listed as >90% trans. The manuscript should specify whether the reported GC measurements were only of the trans isomer. Also, the manuscript should specify the cis/trans composition of 4-hexen-3-one as it appears in nature (if known). Every time the description of the calculations identifies the test compounds, they should specify trans (e.g., "*trans*-4-hexen-3-one" rather than "4-hexen-3-one").

3b.  Is the 5-hexen-2-one used in experiments all trans?

**Response:** 4-hexen-3-one exists in the form of trans (90%) and cis isomers (10%) in nature. Hence, we have used trans-4-hexen-3-one in our experiments. 5-hexen-2-one exists only in one conformation, which was used in present experiments. This is incorporated in the RMS.

R2Q14. In the computational methodology, expand the acronyms CVT and SCT. The partition function of the reactant does not depend on *s*, as stated here. VMEP should be specified as a potential energy **difference** (corrected for zero-point energy). The value of the reaction path degeneracy for each TS should be specified somewhere.

**Response:** Acronyms CVT and SCT are expanded in introduction as Canonical Variational Transition state theory (CVT) with Small Curvature Tunneling (SCT) and have used as CVT and SCT throughout the text.

The partition function of the reactant does not depend on s. Hence, we have edited the sentence as "$\phi^R$ and $Q^{GT}$ are the partition functions of generalized reactant and transition state respectively".

's' is a reaction coordinate parameter that determines the location of the generalized transition dividing surface.

$V_{MEP}(s)$ is the potential along the reaction path at 's' and the minimum energy pathways (MEP) was constructed with a gradient step size of 0.01.

For all the title reactions, all the transitions states are independent and the reaction path degeneracy is one ($\sigma=1$). This is added in the RMS.

R2Q16. In Table 10, specify whether the experimental or theoretical rate constants used for Cl reactions with 4-hexen-3-one and 5-hexene-2-one. Also, add a second digit to the lifetimes with respect to reaction with OH, and only use two significant figures elsewhere.

**Response:** To calculate the lifetimes of test molecules, we have used present experimentally measured rate coefficients. The significant figures are corrected in the RMS as suggested.

R2Q17. Caption to Table 11. These are "percentages" not "ratios".

**Response:** The word "ratios" is replaced with "percentages" in the RMS.

R2Q18. Conformers:

- Did the authors choose the conformers because they were the minimum energy conformers? If so, what efforts were made to verify this?

- Two of the three test compounds are listed as having near-Cs symmetry, and probably should be treated as having Cs symmetry. This means that there are fewer unique transition states than listed (e.g., for 4-hexen-3-one TSs 8-10 are only two unique TSs).

**Response:** We have optimized all the possible conformers during geometry optimization. The lowest energy conformers were considered for rate coefficients calculations. Whereas other possible conformers are more than 1.9 kcal mol$^{-1}$ higher in energy than the lowest energy conformers and therefore, it is unlikely to have significant contribution to the reaction in the temperature range of our study.

In methyl group of the 4-hexen-3-one, three H-abstraction transition states (TS8-TS10) are there. Out of three, two transition states are having the similar energies (*c.f.* Tables 6, 7 and 8). Although they are having similar energies, we have considered all the transition states in the calculation of the total rate coefficients for all three reactions, as they are structurally different. Therefore, it may not be helpful if the molecule is considered to have Cs symmetry.

R2Q19. Can Cl form a van der Waals complex with the □cloud of the carbonyl groups?

**Response:** There are possibilities for the formation of van der Waals complexes for transition states TS2 and TS7 in case of reaction R1; TS2 and TS5 in case of reaction R2; TS2 and TS5 in case of reaction R3. However, when we have optimized all these structures, we did not observe

any van der Waals complexes as the bond lengths are greater than 3.5 Å between Cl and cloud of the carbonyl group.

R2Q20. Both "test" and "sample" are used for the alkenones; please standardize terminology
**Response:** It was corrected and "test" was used throughout the RMS.

R2Q21. In Table 1, some of the error bars don't make sense, e.g., bottom of page 21 (298 K): for 1,3 butadiene as a reference, the error bar on ktest/kref is 25%, but the error bar on ktest is only 7%. for isoprene as a reference, the error on the three individual values of ktest/kref is on the orderof 7-9%, and that should be reflected in the average value of ktest/kref. for the final value of ktest averaged over multiple reference compounds, the error bar should be closer to 1.0 than 0.4 ($\times$ 10-10 cm3 molecule-1 sec-1).
**Response:** We have re calculated errors and a separate section was included on error analysis in the RMS as mentioned below.
"The uncertainties in the temperature (within ±2K) and pressure (within ±1 Torr) in the reaction chamber were very small and did not contribute significantly on the determination of the rate coefficients. The elution of the test molecules and reference compounds in the GC are precise and the uncertainty in concentrations was estimated to be less than 5%. For each experiment, the obtained slopes (using linear least squares method) along with the errors (95% confidence limit) are given in Tables 1 and 2. The uncertainties on the weighted average slopes $((k_{sample}/k_{reference})_{Average})$ are determined using the error propagation method according to the equation: $\Delta y/y = [[\Delta a/a]^2 + [\Delta b/b]^2 + .....]^{1/2}$, where $\Delta y/y$ is the relative error on the average slope and $[\Delta a/a]$, $[\Delta b/b]$ are the relative errors on the individual slopes. The errors quoted for the rate coefficients also include the quoted error in the rate coefficients for the reference reactions and are calculated using the standard error propagation method which was used by several groups (Blanco et al., 2009; Stoeffler et al., 2013; Peirone et al., 2014 and Dash et al., 2015 ) according to the equation: $\Delta k_{test} = k_{test} \times [(\Delta k_{ref}/k_{ref})^2 + (\Delta(k_{test}/k_{ref})/ (k_{test}/k_{ref}))^2]^{1/2}$, where $(\Delta k_{ref}/k_{ref})$ and $\Delta(k_{test}/k_{ref})/(k_{test}/k_{ref})$ are the relative errors on $k_{ref}$ and $k_{test}/k_{ref}$, respectively. At every temperature, the uncertainties in the averaged rate coefficients were calculated according to the equation: $\Delta k_{average} = k_{average} \times [[\Delta l/k_l]^2 + [\Delta m/k_m]^2 + [\Delta n/k_n]^2]^{1/2}$, where $\Delta l$, $\Delta m$ and $\Delta n$, are the relative errors on the individual rate coefficients and $k_l$, $k_m$ and $k_n$ are individual rate coefficients.

A major source of systematic errors in the determination of the title reaction's rate coefficients are from the absolute uncertainties in the rate coefficients of the reference reactions.

Blanco, M. B., Bejan, I., Barnes, I., Wiesen, P., Teruel, M. A. Temperature-dependent rate coefficients for the reactions of Cl atoms with methyl methacrylate, methyl acrylate and butyl methacrylate at atmospheric pressure. Atmos. Environ. 43, 5996–6002, 2009.

Stoeffler, C., Joly, L., Durry, G., Cousin, J., Dumelie, N., Bruyant, A., Roth, E., Chakir, A. Kinetic study of the reaction of chlorine atoms with hydroxyacetone in gas-phase. Chem. Phys. Lett. 590, 221–226, 2013.

Peirone, S. A., Barrera, J. A., Taccone, R. A., Cometto, P. M., Lane, S. I. Relative rate coefficient measurements of OH radical reactions with (Z)-2-hexen-1-ol and (E)-3-hexen-1-ol under simulated atmospheric conditions. Atmos. Environ. 85, 92-98, 2014.

Dash, M. R., Srinivasulu, G., Rajakumar, B. Experimental and computational investigation on the gas phase reaction of p-cymene with Cl atoms. J. Phys. Chem. A 119, 559−570, 2015.

R2Q22. The Supporting Information is more complete than many, but it should also include absolute energies at 0 K, zero-point energies, and (ideally) H and G at 298 K. Also add the CVT rate constant and tunneling corrections versus temperature for each reaction path for all three test molecules.

**Response:** Absolute energies at 0K, zero-point energies and thermodynamic parameters such as enthalpies, Gibbs free energies and entropies were given in the revised supporting information along with CVT rate constants and tunneling corrections versus temperature for each reaction path of each reaction.

R2Q23. The Introduction does not reflect a thorough understanding of atmospheric chemistry and cites too few recent papers.

**Response:** Now, we have added the below given references, where the usage of laboratory studies on atmospheric chemistry were described in detail. The descriptions include atmospherically relevant processes which provides the fundamental information on climate change, urban air pollution, stratospheric ozone depletion and ecosystem health. This is added in the RMS.

Burkholder, J. B., Jonathan, P. D. A., Barnes, I., Roberts, J.M., Melamed, M. L., Ammann, M., Christopher, D.C., Annmarie, G.C., Lucy, J.C. and Crowley, J.N. et al. The essential role for laboratory studies in atmospheric chemistry, Environ. Sci. Technol., 51, 2519-2528, 2017.

Ng, N. L.; Brown, S. S.; Archibald, A. T.; Atlas, E.; R.C, C.; Crowley, J. N.; Day, D. A.; Donahue, N. M.; Fry, J. L.; al, F. Nitrate radicals and biogenic volatile organic compounds: Oxidation, mechanisms and organic aerosol. Atmos. Chem. Phys. 17, 2103-2162, 2017.

Poschl, U.; Shiraiwa, M. Multiphase chemistry at the atmosphere-biosphere interface influencing climate and public health in the anthropocene. Chem. Rev. 115, 4440−4475, 2015.

West, J. J.; Cohen, A.; Dentener, F.; Brunekreef, B.; Zhu, T.; Armstrong, B.; Bell, M. L.; Brauer, M.; Carmichael, G.; Costa, D. L.; et al. What we breathe impacts our health: Improving understanding of the link between air pollution and health. Environ. Sci. Technol. 50, 4895−4904, 2016.

Liggio, J.; Li, S. M.; Hayden, K.; Taha, Y. M.; Stroud, C.; Darlington, A.; Drollette, B. D.; Gordon, M.; Lee, P.; Liu, P.; et al. Oil sands operations as a large source of secondary organic aerosols. Nature 534, 91−95, 2016.

R2Q24. Page 6: use a lower case rather than upper case kappa for tunneling corrections.
Response: Now lower case Kappa is used in the RMS.

R2Q25. The equations given for rate constants on page 6 have units of sec-1. Please correct them.
Response: corrected.

R2Q26. On page 7, the results of Bedjanian et al. were at low pressure, not atmospheric pressure, although the rate constant was reported to be independent of pressure.
Response: It was corrected in the RMS as given below.
Bedjanian et al. have reported the temperature dependent rate coefficient for the reaction of isoprene with Cl atom in the temperature range of 233-320K and at low pressure.

R2Q27. On page 7, lines 28 "10%" should be "16%"

Response: Corrected.